# Control of meiotic entry by dual inhibition of a key mitotic transcription factor

Elçin Ünal*, Amanda J Su, Siri C Yendluri

Department of Molecular and Cell Biology, University of California, Berkeley, Berkeley, United States

**Abstract** The mitosis to meiosis transition requires dynamic changes in gene expression, but whether and how the mitotic transcriptional machinery is regulated during this transition is unknown. In budding yeast, SBF and MBF transcription factors initiate the mitotic gene expression program. Here, we report two mechanisms that work together to restrict SBF activity during meiotic entry: repression of the SBF-specific Swi4 subunit through LUTI-based regulation and inhibition of SBF by Whi5, a functional homolog of the Rb tumor suppressor. We find that untimely SBF activation causes downregulation of early meiotic genes and delays meiotic entry. These defects are largely driven by the SBF-target G1 cyclins, which block the interaction between the central meiotic regulator Ime1 and its cofactor Ume6. Our study provides insight into the role of $SWI4^{LUTI}$ in establishing the meiotic transcriptional program and demonstrates how the LUTI-based regulation is integrated into a larger regulatory network to ensure timely SBF activity.

## eLife assessment

This study highlights several **important** regulatory pathways that contribute to the control of entry into meiosis by turning down mitotic functions. Central to this regulation is the control of Swi4 level and activity, and **convincing** overexpression experiments identify downstream effectors of Swi4.

*For correspondence:
elcin@berkeley.edu

## Introduction

A key aspect in understanding developmental programs and cell state transitions is mapping the interplay between transcription factors and their associated gene regulatory networks. In the budding yeast *Saccharomyces cerevisiae*, the transition from mitotic growth to meiotic differentiation is a crucial decision that is regulated by multiple inputs, such as nutrient availability, respiration competence, and cell identity. Under nutrient-limiting conditions, a diploid cell enters the meiotic program to produce four haploid gametes. The process of meiotic entry is tightly controlled by the master transcriptional regulator Ime1, as both extrinsic (e.g. nutrient status, extracellular pH) and intrinsic (e.g. mating type, mitochondrial function) cues are integrated at the *IME1* promoter (**Kassir et al., 1988**; **Honigberg and Purnapatre, 2003**; reviewed in **van Werven and Amon, 2011**). Once translated, Ime1 is phosphorylated by the Rim11 and Rim15 kinases to promote its nuclear localization and interaction with Ume6 (**Vidan and Mitchell, 1997**; **Pnueli et al., 2004**; **Malathi et al., 1999**; **Malathi et al., 1997**). In mitotically dividing cells, Ime1 target genes are repressed by Ume6 through its association with the Sin3-Rpd3 histone deacetylase complex (**Kadosh and Struhl, 1997**; **Rundlett et al., 1998**). However, under nutrient starvation, entry into the meiotic program is initiated by the interaction between Ime1 and Ume6, which together function as a transcriptional activator, culminating in the induction of early meiotic genes (**Bowdish et al., 1995**; **Harris and Ünal, 2023**). Mitosis to meiosis transition requires dynamic remodeling of the gene regulatory networks to maintain the mutual exclusivity of these programs. While entry into the mitotic program is initiated by the central transcription factors SBF

and MBF (*Spellman et al., 1998*; *Iyer et al., 2001*), whether and how these complexes are regulated during meiotic entry is unknown.

The molecular mechanisms regulating entry into the mitotic cell cycle, also known as G1/S transition, are functionally conserved from yeast to metazoans (reviewed in *van den Heuvel and Dyson, 2008*). SBF and MBF are heterodimeric transcription factors composed of Swi4-Swi6 and Mbp1-Swi6 subunits, respectively (*Figure 1A*). Although there is no sequence homology, these transcription factors are functionally homologous to the mammalian E2Fs. E2Fs are negatively regulated by the tumor suppressor protein Rb, which is homologous to the budding yeast Whi5 that inhibits SBF in early G1 (*de Bruin et al., 2004*; *Costanzo et al., 2004*; *Hasan et al., 2014*). Whi5-based SBF inhibition is relieved by cyclin/CDK-dependent phosphorylation and subsequent re-localization of Whi5 from the nucleus to the cytoplasm (*Wagner et al., 2009*). Although SBF and MBF act in parallel during the G1/S transition (*Spellman et al., 1998*; *Iyer et al., 2001*), they activate functionally specialized subsets of gene targets. SBF regulates the expression of genes involved in budding and cell morphogenesis, while MBF-regulated genes are involved in DNA replication and repair (*Iyer et al., 2001*; *Simon et al., 2001*). Despite the well-characterized function in budding yeast mitotic growth, the regulation and function of SBF and MBF during meiotic entry remains largely unknown. In contrast to mitotic divisions where budding and DNA replication occur simultaneously, the meiotic program requires DNA replication and repair, but not bud morphogenesis or asymmetric growth. Accordingly, the regulation and subsequent activity of SBF and MBF are likely to be divergent to establish meiotic entry.

The activity of SBF and MBF is regulated in part through subunit abundance, which in turn controls expression of the G1/S regulon (*Dorsey et al., 2018*). For example, overexpression of a hyperactive allele of *SWI4* can trigger premature entry into the cell cycle (*Sidorova and Breeden, 2002*). Additionally, *SWI4* has also been shown to be haploinsufficient and rate limiting during G1/S progression (*McInerny et al., 1997*). These results suggest that the precise levels of SBF and MBF subunits are important for activating the G1/S regulon at the correct time.

Three key observations support differential regulation of SBF and MBF during the meiotic program: First, a meiotic mass spectrometry dataset suggests that Swi4 has dynamic protein behavior, which is not observed for its counterparts Swi6 and Mbp1 (*Cheng et al., 2018*). Second, meiotic cells express a non-canonical mRNA from the *SWI4* locus called LUTI (*Brar et al., 2012*; *Tresenrider et al., 2021*), which stands for Long Undecoded Transcript Isoform (*Chen et al., 2017*; *Chia et al., 2017*). Third, two SBF-specific targets, namely the G1-cyclins *CLN1* and *CLN2*, have been shown to repress early meiotic gene expression (*Colomina et al., 1999*).

LUTI-based gene regulation repurposes gene-activating transcription factors as repressors by a two-pronged mechanism (*Chen et al., 2017*; *Chia et al., 2017*; *Tresenrider and Ünal, 2018*; *Tresenrider et al., 2021*). First, transcription factor-dependent activation of the LUTI promoter results in co-transcriptional silencing of the downstream canonical gene promoter. Second, the coding sequence (CDS) within the LUTI is translationally repressed due to competitive translation of the upstream open reading frames (uORFs) in the LUTI-specific 5′ leader. Consequently, upregulation of the LUTI results in downregulation of the canonical mRNA and corresponding protein. While LUTIs are conserved from yeast to humans and have been identified in different cellular contexts (*Cheng et al., 2018*; *Van Dalfsen et al., 2018*; *Hollerer et al., 2019*; *Jorgensen et al., 2020*), their biological significance remains poorly understood. In fact, only a single LUTI has been assigned a biological role so far with distinct phenotypic outcomes resulting from its loss or gain of function (*Chen et al., 2017*; *Chia et al., 2017*).

In this study, we aimed to investigate the differential regulation of SBF and MBF during the meiotic program and determine how the untimely activation of SBF impacts meiotic entry. We found that overexpression of Swi4 results in the activation of SBF targets and concomitant downregulation of the early meiotic genes. SBF targets Cln1 and Cln2 inhibit meiotic entry by blocking the interaction between Ime1 and its cofactor Ume6. Furthermore, the LUTI-based mechanism causes downregulation of Swi4 protein synthesis and acts in conjunction with Whi5 to restrict SBF activity during meiosis. Overall, our study reveals the functional role of a LUTI in establishing the early meiotic transcriptome, demonstrates how the LUTI-based regulation is integrated into a larger regulatory network to ensure timely SBF activity, and provides mechanistic insights into how SBF misregulation impedes transition from mitotic to meiotic cell fate.

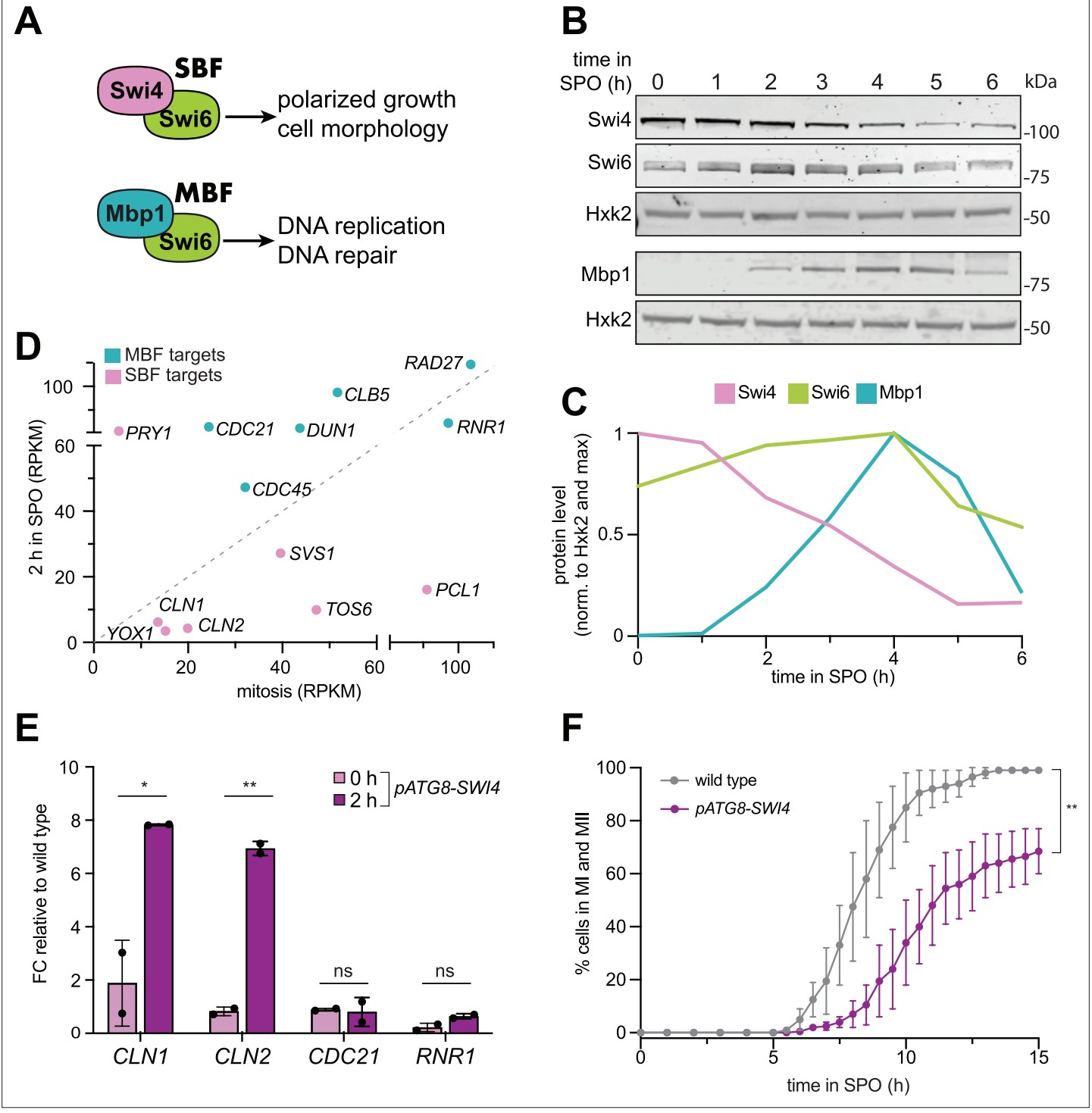

**Figure 1.** Swi4 subunit of the SBF and SBF targets are downregulated during early meiosis. (**A**) A schematic of SBF and MBF complexes and the general functional groups of the genes they regulate. (**B**) Samples from strain UB35246 were collected between 0 and 6 hr (h) in sporulation medium (SPO) and immunoblots were performed using α-Swi4, α-Swi6, and α-Mbp1 respectively. Hxk2 was used a loading control. Representative blots from one of two biological replicates are shown. (**C**) Quantification of the immunoblots in (**B**). The signal at each time point was first normalized to Hxk2 loading control and then to the max signal. (**D**) Scatterplot of RNA-seq data (RPKM) from ***Brar et al., 2012*** comparing 2 hr in SPO vs. mitotic growth of well characterized SBF targets (pink) and MBF targets (teal). (**E**) Wild type (UB22199) and *pATG8-SWI4* (UB22226) cells were collected to perform RT-qPCR for *CLN1, CLN2, CDC21*, and *RNR1* transcripts. Transcript abundance was quantified using primer sets specific for each respective gene from three technical replicates for each biological replicate. Quantification was performed in reference to *PFY1* and then normalized to wild-type control. FC=fold change. Experiments were performed twice using biological replicates, mean value plotted with range. Differences in wild type versus *pATG8-*

*Figure 1 continued on next page*

*Figure 1 continued*

*SWI4* transcript levels at 2 hr in SPO compared with a two-tailed t-test (*, p=0.0351 [*CLN1*]; **, p=0.0013 [*CLN2*]; ns, p=0.8488 [*CDC21*]; ns, p=0.0859 [*RNR1*]). (**F**) Live-cell imaging of strains containing the fluorescently tagged histone Htb1-mCherry for wild type (UB32085) and *pATG8-SWI4* (UB32089). Experiments were performed twice using biological replicates, mean value plotted with range. Differences in meiotic progression tested by Mann Whitney test, two-tailed (**, p=0.0045).

The online version of this article includes the following source data and figure supplement(s) for figure 1:

**Source data 1.** Original file for the immunoblot shown in *Figure 1B* (anti-Swi4, anti-Swi6, anti-Mbp1, anti-Hxk2).

**Source data 2.** Original file for the immunoblot shown in *Figure 1B* with highlighted bands and sample labels (anti-Swi4, anti-Swi6, anti-Mbp1, anti-Hxk2).

**Figure supplement 1.** Swi4 protein abundance in wild-type and *pATG8-SWI4* cells.

**Figure supplement 1—source data 1.** Original file for the immunoblot shown in *Figure 1—figure supplement 1* (anti-V5 [for detecting Swi4-3V5], anti-Hxk2).

**Figure supplement 1—source data 2.** Original file for the immunoblot shown in *Figure 1—figure supplement 1* with highlighted bands and sample labels (anti-V5 [for detecting Swi4-3V5], anti-Hxk2).

## Results

### Swi4 is the sole downregulated subunit within the SBF and MBF complexes during meiosis

While the SBF and MBF transcription factors have been heavily studied in the context of the mitotic cell cycle, their involvement in regulating the meiotic transcriptional program is not well understood. To understand the regulation and function of these complexes during meiosis, we first monitored the levels of SBF and MBF subunits throughout a meiotic time course. Cells were first grown in rich media overnight and were then transferred to pre-sporulation media. After additional overnight growth, cells were shifted to sporulation media (SPO) to induce meiosis, and samples were taken hourly for protein extraction and immunoblotting to monitor the abundance of each subunit. Unlike the mitotic G1/S transition (*Kelliher et al., 2018*), meiotic entry resulted in ~30% decrease in Swi4 levels after 2 hr, while Mbp1 and Swi6 levels were increased (*Figure 1B and C*). These data indicate that Swi4 is the sole subunit within SBF and MBF whose level declines during meiotic entry and are consistent with a published mass spectrometry dataset (*Cheng et al., 2018*).

We next examined the expression of a set of well-characterized transcripts that are regulated by SBF or MBF (*Iyer et al., 2001*; *Simon et al., 2001*; *Bean et al., 2005*; *Harris et al., 2013*; *Smolka et al., 2012*) using a published meiotic RNA-seq dataset (*Brar et al., 2012*). This analysis revealed that compared to mitotically dividing cells, most of the SBF-specific target genes had either low or no expression during early meiosis (*Figure 1D*). In contrast, most of the MBF targets displayed increased expression upon meiotic entry (*Figure 1D*).

Based on our findings thus far, we propose that Swi4 levels are downregulated in meiosis to ensure that SBF targets, including *CLN1* and *CLN2*, are turned off during meiotic entry. If so, then higher Swi4 levels should lead to increased expression of SBF targets. To test this possibility, we overexpressed *SWI4* by placing it under the regulation of the *ATG8* promoter (*pATG8-SWI4*), which is highly expressed in meiosis (*Brar et al., 2012*). The steady-state level of *pATG8*-driven Swi4 was five times higher than wild type (*Figure 1—figure supplement 1*). Using reverse transcription coupled with quantitative polymerase chain reaction (RT-qPCR), we measured the transcript levels of well-characterized SBF targets and observed a significant increase in the expression of *CLN1* and *CLN2* in *pATG8-SWI4* cells relative to wild type (*Figure 1E*, p=0.0351 [*CLN1*], p=0.0013 [*CLN2*], two-tailed t-test). In contrast, the MBF-specific targets *CDC21* and *RNR1* remained similar (*Figure 1E*, p=0.8488 [*CDC21*], p=0.0859 [*RNR1*], two-tailed t-test). These findings indicate that upregulation of *SWI4* is sufficient to induce the expression of SBF-specific targets *CLN1* and *CLN2* in meiosis without affecting MBF-specific targets.

To test whether the downregulation of *SWI4* is functionally important for meiotic progression, we used time-lapse fluorescence microscopy and visualized the kinetics of meiotic divisions in *pATG8-SWI4* cells relative to wild type. By tracking the endogenous histone H2B fused to the red fluorescent protein mCherry (Htb1-mCherry), we found that *SWI4* overexpression caused a significant delay in meiotic progression (*Figure 1F*, p=0.0045, Mann-Whitney test). We conclude that downregulation of *SWI4* is necessary for timely meiotic progression.

## Regulation of Swi4 abundance is required for timely meiotic entry

Since Swi4 abundance, and by inference SBF activity, was decreased during the mitosis-to-meiosis transition, we hypothesized that the meiotic progression delay observed in *pATG8-SWI4* cells was due to meiotic entry defects. To test this we monitored Ime1, a meiosis-specific transcription factor and a master regulator of meiotic entry (*Kassir et al., 1988*; *Honigberg and Purnapatre, 2003*; reviewed in *van Werven and Amon, 2011*). To quantify the bulk levels of Ime1 protein in meiosis, we performed immunoblotting. During meiotic entry (2 hr in SPO), when Swi4 abundance was elevated, there was a 50% reduction in Ime1 levels in *pATG8-SWI4* cells compared to wild type (*Figure 2—figure supplement 1*). In parallel, we monitored meiotic entry on a single-cell basis by measuring the localization of endogenous Ime1 carrying an N-terminal green fluorescent protein tag (*GFP-IME1*; *Moretto et al., 2018*) and Htb1-mCherry. Compared to wild type where >90% of the cells had nuclear Ime1 following meiotic entry (2 hr in SPO), only ~50% of *pATG8-SWI4* cells had nuclear Ime1, which was significantly lower (*Figure 2A and B*, p=0.0169, two-tailed t-test).

Given that the increase in Swi4 levels coincided with a decrease in Ime1 protein expression and nuclear localization, this raised the possibility that Swi4 interferes with Ime1 function. To further investigate the relationship between these two transcription factors at a single-cell level, we generated strains carrying endogenous fluorescent protein tags for each transcription factor (*GFP-IME1, SWI4-mCherry*). Using DAPI as a nuclear marker, we measured the mean nuclear intensity of each transcription factor before (0 hr) and after meiotic entry (4 hr) (see Materials and methods for details on image quantification). In wild type, most cells exhibited decreased nuclear Swi4 and increased nuclear Ime1 upon meiotic entry (*Figure 2C–E*). Conversely, in the *pATG8-SWI4* mutant, there was a significant shift in the fraction of cells with higher nuclear Swi4 levels (*Figure 2D and E*, p<0.0001, Mann-Whitney test,). Additionally, the cells with increased nuclear Swi4 had reduced levels of nuclear Ime1 (*Figure 2E*, p<0.0001, Mann-Whitney test). This inverse relationship between Swi4 and Ime1 nuclear localization further indicates that higher levels of Swi4 are antagonistic to Ime1 function.

Ime1 is necessary for the expression of early meiotic genes. Therefore, we next characterized the changes in the early meiotic transcriptome upon *SWI4* overexpression by mRNA-sequencing (mRNA-seq). Using DESeq2 (*Love et al., 2014*), we identified differentially expressed genes in *pATG8-SWI4* mutant compared to wild type during meiotic entry (2 hr in SPO). To investigate general pathways being affected by increased Swi4 levels in meiosis, we ran gene ontology analysis of statistically significant (padj <0.05) output from DESeq2, which revealed that the genes with significantly increased expression were involved in mitosis while those with significantly decreased expression were involved in meiotic processes (*Figure 2—figure supplement 2*). Regarding the SBF targets, we noticed increased expression of many genes, including *CLN1* and *CLN2* (*Figure 2F*, *Supplementary file 1*). We focused on a cluster of previously identified early meiotic genes involved in DNA replication and recombination (*Brar et al., 2012*) and observed that more than 50% had a significant decrease in their expression upon *SWI4* overexpression (padj <0.05, *Figure 2F*, *Supplementary file 1*). Finally, gene set enrichment analysis was performed to test whether expression of either the early meiotic gene set or SBF target gene set is enriched in wild type or *pATG8-SWI4* mutant cells (*Subramanian et al., 2005*). This gene set level analysis revealed significant enrichment of SBF target gene expression (NES=2.88, p<0.001, *Figure 2G*), as well as significant disenrichment of early meiotic gene expression (NES=–2.88, p<0.001, *Figure 2G*) in the *pATG8-SWI4* mutant. Altogether, these findings demonstrate that the increased levels of Swi4 during transition from mitotic to meiotic cell fate abruptly activates SBF and disrupts the early meiotic transcriptome, highlighting the importance of *SWI4* regulation.

As an orthogonal approach, we performed live-cell imaging of Rec8, endogenously tagged with GFP (Rec8-GFP), which is a meiosis-specific cohesin subunit and a direct transcriptional target of Ime1 (*Primig et al., 2000*; *Klein et al., 1999*). Htb1-mCherry was used as a nuclear marker. This analysis revealed a significant delay in Rec8-GFP nuclear appearance in the *pATG8-SWI4* strain compared to wild type (*Figure 2H and I*, p=0.0005, Mann-Whitney test). Furthermore, sporulation efficiency was decreased by ~20% in the *pATG8-SWI4* strain relative to wild type (*Supplementary file 2*). These findings are consistent with the mRNA-seq data and further underscore the importance of *SWI4* downregulation in establishing a robust meiotic cell fate.

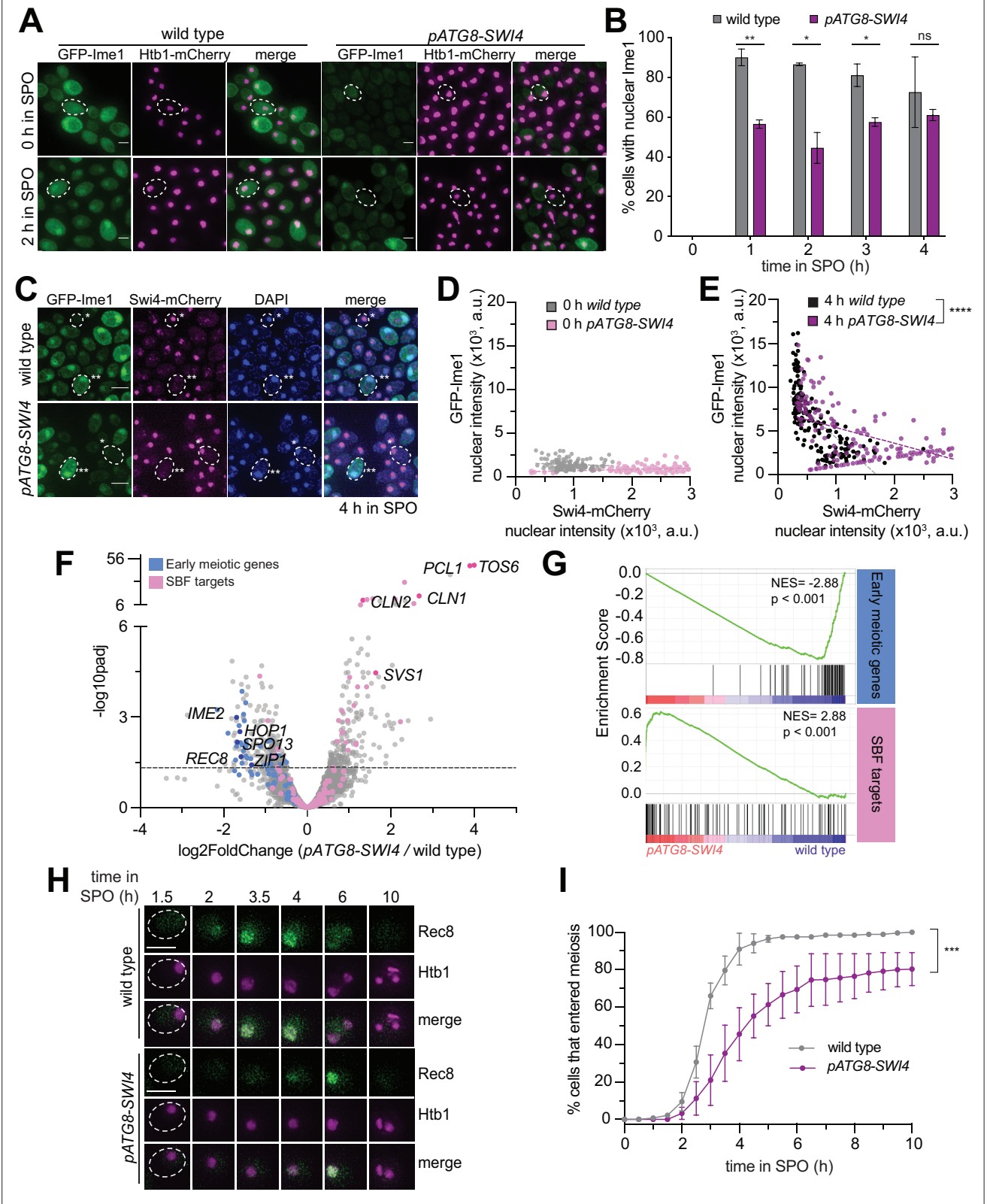

**Figure 2.** Regulation of Swi4 abundance is required for timely meiotic entry. (**A–B**) Fixed imaging of cells marked with GFP-Ime1 and Htb1-mCherry. Wild type (UB22199) and *pATG8-SWI4* (UB22226) cells were collected between 0 and 4 hr in SPO. (**A**) Representative images with merge at 0 hr and 2 hr in SPO. Representative cells outlined. Scale bar: 3μm. (**B**) Quantification as percent cells with nuclear Ime1. Experiments were performed twice using biological replicates, mean value plotted with range. Total of 200 cells analyzed per strain. Differences in the fraction of cells with nuclear Ime1

*Figure 2 continued on next page*

*Figure 2 continued*

was compared using a two-tailed t-test (\*\*, p=0.0099 [1 hr in SPO]; \*, p=0.0169 [2 hr in SPO]; \*, p=0.0315 [3 hr in SPO]; ns, two-tailed p=0.4595 [4 hr in SPO]). (**C–E**) Fixed imaging of cells marked with GFP-Ime1 and Swi4-mCherry. Wild type (UB31378) and *pATG8-SWI4* (UB31381) cells were collected at 0 hr and 4 hr in SPO. (**C**) Example images with merge at 4 hr in SPO. Example cells outlined (\*low nuclear GFP-Ime1 with high nuclear Swi4-mCherry, \*\*high nuclear GFP-Ime1 with low nuclear Swi4-mCherry). Scale bar: 3 µm. (**D**) Scatterplot of GFP-Ime1 mean nuclear intensity and Swi4-mCherry mean nuclear intensity for wild type and *pATG8-SWI4* cells at 0 hr in SPO. See Materials and methods for further details about image quantification. Dashed line is linear regression plotted for each condition and strain. A total number of 269 cells were analyzed. (**E**) Same as in (**D**) but for wild type and. *pATG8-SWI4* cells at 4 hr in SPO. A total number of 341 cells were analyzed. Differences in mean nuclear GFP-Ime1 or Swi4-mCherry intensity between wild type and *pATG8-SWI4* compared using a Mann-Whitney test, two-tailed (\*\*\*\*, p<0.0001 [wild type vs. *pATG8-SWI4* (GFP-Ime1)]; \*\*\*\*, p<0.0001 [wild type vs. *pATG8-SWI4* (Swi4-mCherry)]). (**F**) Volcano plot of DE-Seq2 analysis for *pATG8-SWI4* versus wild type. Dashed line indicates padj (p value)=0.05. Analysis was performed using mRNA-seq from two biological replicates. Wild type (UB22199) and *pATG8-SWI4* (UB22226) cells were collected at 2 hr in SPO. SBF targets (pink) (*Iyer et al., 2001*) and early meiotic genes (blue) defined by *Brar et al., 2012*. Darker pink or darker blue, labeled dots are well studied targets in either gene set list. (**G**) GSEA analysis of mRNA-seq comparing wild type vs. *pATG8-SWI4* collected at 2 hr in SPO. Vertical black bars represent the early meiotic cluster from *Brar et al., 2012* or SBF cluster from *Iyer et al., 2001*. The heatmap indicates genes that are more enriched in *pATG8-SWI4* (red, left-side) or genes that are enriched in wild type (blue, right-side). NES=normalized enrichment score. Enrichment was determined by comparing *pATG8-SWI4* versus wild type. (**H–I**) Live-cell imaging of cells in meiosis marked by Rec8-GFP and nuclear marker Htb1-mCherry for wild type (UB32085) and *pATG8-SWI4* (UB32089). (**H**) Movie montage with example images throughout meiosis for Rec8-GFP and Htb1-mCherry. Scale bar: 3 µm. (**I**) Quantification as percent of cells that entered meiosis assayed by nuclear Rec8 appearance. Experiments were performed using two biological replicates, mean value plotted with range. A total number of 452 cells were analyzed. Differences in meiotic progression compared by Mann Whitney test, two-tailed (\*\*\*, p=0.0005 [wild type vs. *pATG8-SWI4*]).

The online version of this article includes the following source data and figure supplement(s) for figure 2:

**Figure supplement 1.** Ime1 protein abundance in wild-type and *pATG8-SWI4* cells.

**Figure supplement 1—source data 1.** Original file for the immunoblot shown in *Figure 2—figure supplement 1* (anti-GFP [for detecting GFP-Ime1], anti-Hxk2).

**Figure supplement 1—source data 2.** Original file for the immunoblot shown in *Figure 2—figure supplement 1* with highlighted bands and sample labels (anti-GFP [for detecting GFP-Ime1], anti-Hxk2).

**Figure supplement 2.** Gene ontology analysis for mRNA-seq comparing *pATG8-SWI4* to wild type.

## Removal of the SBF targets Cln1 or Cln2 partially rescues the meiotic entry delay in *pATG8-SWI4* mutants

Overexpression of the G1 cyclins *CLN1*, *CLN2*, and *CLN3* has been previously shown to inhibit meiotic entry (*Colomina et al., 1999*). Given that *CLN1* and *CLN2* are transcriptional targets of SBF, we next determined whether the meiotic progression delay in *pATG8-SWI4* cells could be due to increased G1 cyclin protein levels. We first examined Cln1 and Cln2 protein levels using epitope-tagged alleles at their endogenous loci (*CLN1-3V5* or *CLN2-3V5*) in wild-type and *pATG8-SWI4* cells. In response to *SWI4* overexpression, we observed up to twofold and tenfold increase in Cln1 and Cln2 protein levels, respectively, corroborating the mRNA-seq data (*Figure 3A–D*).

To determine whether these two G1 cyclins are functionally responsible for the meiotic progression delay observed in response to SBF misregulation, we performed time-lapse fluorescence microscopy in cells carrying Rec8-GFP and Htb1-mCherry. We found that deletion of either *CLN1* or *CLN2* significantly rescued the meiotic progression delay in the *pATG8-SWI4* mutant (p=0.0111 [*cln1Δ*], p=0.0478 [*cln2Δ*], Mann-Whitney test) (*Figure 3E and F*). The kinetics of meiotic entry in the absence of either Cln1 or Cln2 closely resembled that of wild type (*Figure 3—figure supplement 1*), indicating that the observed rescue was specific to the meiotic defect resulting from elevated Swi4 levels. However, *pATG8-SWI4* cells lacking either *CLN1* or *CLN2* were still delayed compared to wild type, suggesting redundancy. We were unable to examine meiotic progression in the *cln1Δ cln2Δ* double mutant due to its severe sickness in the SK1 background. Nevertheless, our analyses establish a causal link demonstrating that both *CLN1* and *CLN2* contribute to the meiotic defects arising from SBF misregulation.

## Tethering of Ime1 to Ume6 is sufficient to overcome the meiotic block exerted by G1 cyclin overexpression

Given the partial rescue of the meiotic delay in *pATG8-SWI4* mutants by the deletion of individual G1 cyclins, we next investigated how the G1 cyclins interfere with meiosis. To this end, we generated transgenes that expressed either *CLN1* or *CLN2* under the control of the *pATG8* promoter for meiotic overexpression. While wild-type cells successfully completed meiosis with more than 94% sporulation

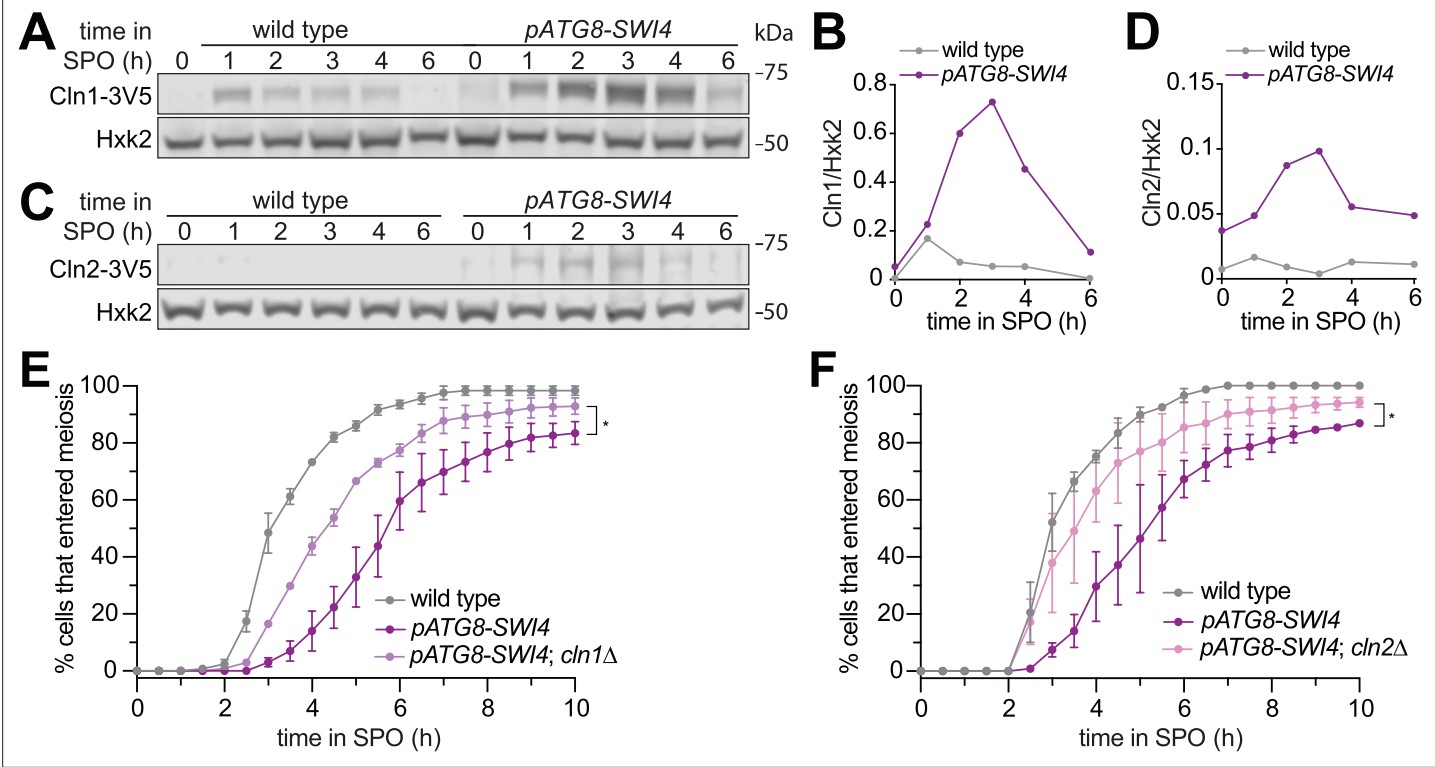

**Figure 3.** Removal of the SBF targets Cln1 or Cln2 partially rescues the meiotic entry delay in the *pATG8-SWI4* mutant. (**A**) Immunoblotting was performed on samples collected for wild type (UB29326) and *pATG8-SWI4* (UB29328) between 0 and 6 hr in SPO using α-V5 antibody to track Cln1-3V5. Hxk2 was used a loading control. Representative blots from one of two biological replicates are shown. (**B**) Quantification of (**A**). (**C**) Same as in (**A**) but for wild type (UB29330) and *pATG8-SWI4* (UB29332) cells using α-V5 antibody to track Cln2-3V5. Hxk2 was used a loading control. Representative blots from one of two biological replicates are shown. (**D**) Quantification of (**C**). (**E**) Live-cell imaging of meiotic cells marked by Rec8-GFP and nuclear marker Htb1-mCherry, with the following genotypes: wild type (UB32085), *pATG8-SWI4* (UB32089), and *pATG8-SWI4; cln1Δ* (UB34536). Quantification of cells that entered meiosis assayed by the initial timing of nuclear Rec8 appearance. Experiments were performed using two biological replicates, mean value plotted with range. A total number of 883 cells were analyzed. Differences in meiotic progression compared by Mann-Whitney test, two-tailed (*, p=0.0111 [*pATG8-SWI4* vs. *pATG8-SWI4; cln1Δ*]). *cln1Δ* alone (not shown) has similar meiotic progression kinetics relative to wild type. (**F**) Same as (**E**) but with the following genotypes: wild type (UB32085), *pATG8-SWI4* (UB32089), and *pATG8-SWI4; cln2Δ* (UB34165). A total number of 610 cells were analyzed. Differences in meiotic progression compared by Mann-Whitney test, two-tailed (*, p=0.0478 [*pATG8-SWI4* vs. *pATG8-SWI4; cln2Δ*]). *cln2Δ* alone (not shown) has similar meiotic progression kinetics relative to wild type.

The online version of this article includes the following source data and figure supplement(s) for figure 3:

**Source data 1.** Original file for the immunoblot shown in *Figure 3A* (anti-V5 [for detecting Cln1-3V5], anti-Hxk2).

**Source data 2.** Original file for the immunoblot shown in *Figure 3A* with highlighted bands and sample labels (anti-V5 [for detecting Cln1-3V5], anti-Hxk2).

**Source data 3.** Original file for the immunoblot shown in *Figure 3C* (anti-V5 [for detecting Cln2-3V5], anti-Hxk2).

**Source data 4.** Original file for the immunoblot shown in *Figure 3C* with highlighted bands and sample labels (anti-V5 [for detecting Cln2-3V5], anti-Hxk2).

**Figure supplement 1.** Meiotic entry upon removal of either Cln1 or Cln2.

efficiency, *CLN2* overexpression resulted in only 8.5% of cells forming gametes. Overexpression of *CLN1* also resulted in a meiotic defect, albeit to a lesser extent than the *pATG8-CLN2* mutant (65% sporulation efficiency, *Figure 4A*).

Since the meiotic defect was more pronounced in response to *CLN2* overexpression, we decided to use the *pATG8-CLN2* mutant to explore how G1 cyclins counteract meiosis. To assess meiotic entry, we performed fixed-cell imaging for GFP-Ime1 and found that when *CLN2* was overexpressed, only 5% of the *pATG8-CLN2* cells displayed nuclear GFP-Ime1 signal during early meiosis as opposed to >84% of wild-type cells (*Figure 4B and C*, 2 hr in SPO). In parallel, we measured *IME1* transcript and Ime1 protein levels. In both cases, we observed ~30% decrease in abundance in the *pATG8-CLN2*

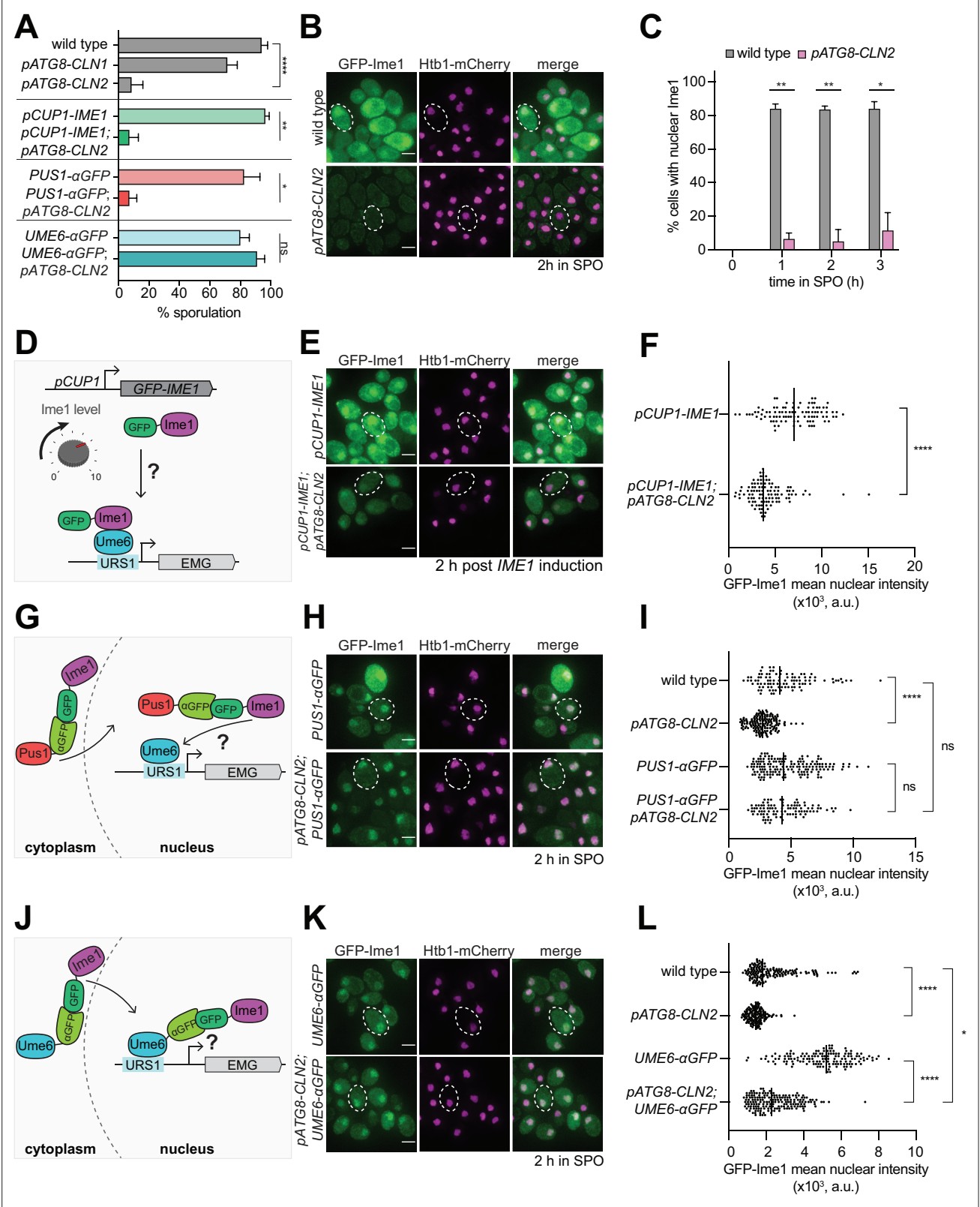

**Figure 4.** Tethering of Ime1 to Ume6 is sufficient to overcome the meiotic block exerted by G1 cyclin overexpression. (**A**) Sporulation efficiency of cells at 24 hr in SPO media wild type (UB22199), *pATG8-CLN1* (UB32820), *pATG8-CLN2* (UB25959), *pCUP-GFP-IME1* (UB34641), *pCUP1-GFP-IME1; pATG8-CLN2* (UB35057), *PUS1-αGFP* (UB35593), *PUS1-αGFP; pATG8-CLN2* (UB35982), *UME6-αGFP* (UB35300), and *UME6-αGFP; pATG8-CLN2* (UB35177). Experiments shown in this figure were performed using two biological replicates, mean value plotted with range. Total of 200 cells counted per strain.

*Figure 4 continued on next page*

*Figure 4 continued*

See ***Supplementary file 2*** for statistics. (**B–C**) Fixed imaging of cells marked with GFP-Ime1 and Htb1-mCherry. Wild type (UB22199) and *pATG8-CLN2* (UB25959) cells were collected between 0 and 3 hr in SPO. (**B**) Representative images with merge at 2 hr in SPO. Representative cells outlined. Scale bar: 3 μm. (**C**) Quantification of cells with nuclear Ime1. Experiments were performed using two biological replicates, mean value plotted with range. Total of 200 cells analyzed per strain. Differences in percent cells with nuclear Ime1 was compared by two-tailed t-test (**, p=0.00917 [1 hr in SPO]; **, p=0.0044 [2 hr in SPO]; *, p=0.0122 [3 hr in SPO]). (**D**) Schematic depicting use of *pCUP1* promoter (*pCUP1-GFP-IME1*) to rescue Ime1 transcript and protein levels. (**E–F**) Fixed imaging of cells marked with GFP-Ime1 and Htb1-mCherry. Cells with the following genotypes were collected at 2 hr in SPO: wild type (UB22199), *pATG8-CLN2* (UB35106), *pCUP1-GFP-IME1* (UB34641), and *pCUP1-GFP-IME1; pATG8-CLN2* (UB35057). (**E**) Representative images with merge and representative cells outlined. Scale bar: 3 μm. (**F**) GFP-Ime1 mean nuclear intensity measured for a single z-slice. A total number of 433 cells were analyzed. Differences in mean nuclear intensity compared by Mann-Whitney test, two tailed, (****, p<0.0001 [*pCUP1-IME1* vs. *pCUP1-IME1; pATG8-CLN2*]). (**G**) Schematic of nanobody trap strategy with *PUS1-αGFP* and GFP-Ime1 to rescue Ime1 nuclear localization in meiosis. (**H–I**) Fixed imaging of cells marked with GFP-Ime1 and Htb1-mCherry. Cells with the following genotypes were collected at 2 hr in SPO: wild type (UB22199), *pATG8-CLN2* (UB35106), *PUS1-αGFP* (UB35593), and *PUS1-αGFP; pATG8-CLN2* (UB35982). (**H**) Representative images with merge and example cells outlined. Scale bar: 3 μm. (**I**) GFP-Ime1 mean nuclear intensity measured for a single z-slice. A total number of 934 cells were analyzed. Differences in mean nuclear intensity compared by Mann-Whitney test, two-tailed, (****, p<0.0001 [wild type vs. *pATG8-CLN2*]; not significant (ns), p=0.6563 [*PUS1-αGFP* vs. *pATG8-CLN2; PUS1-αGFP*]; not significant (ns), p=0.8881 [wildtype vs. *pATG8-CLN2; PUS1-αGFP*]). (**J**) Schematic of nanobody trap strategy with *UME6-αGFP* and GFP-Ime1 to rescue Ime1-Ume6 interaction in meiosis. (**K–L**) Fixed imaging of cells marked with GFP-Ime1 and Htb1-mCherry. Cells with the following genotypes were collected at 2 hr in SPO: wild type (UB22199), *pATG8-SWI4* (UB35106), *UME6-αGFP* (UB35300), and *UME6-αGFP; pATG8-CLN2* (UB35177). (**K**) Representative images with merge and representative cells outlined. Scale bar: 3 μm. (**L**) GFP-Ime1 mean nuclear intensity measured for a single z-slice. A total number of 1220 cells were analyzed. Differences in mean nuclear intensity compared by Mann-Whitney test, two-tailed (****, p<0.0001 [wild type vs. *pATG8-CLN2*]; ****, p<0.0001 [*UME6-αGFP* vs. *pATG8-CLN2; UME6-αGFP*]; *, p=0.0354 [wildtype vs. *pATG8-CLN2; UME6-αGFP*]).

The online version of this article includes the following source data and figure supplement(s) for figure 4:

**Figure supplement 1.** *IME1* transcript levels upon C*LN2* overexpression.

**Figure supplement 2.** Ime1 protein levels upon C*LN2* overexpression.

**Figure supplement 2—source data 1.** Original file for the immunoblot shown in ***Figure 4—figure supplement 2*** (anti-GFP [for detecting GFP-Ime1], anti-Hxk2).

**Figure supplement 2—source data 2.** Original file for the immunoblot shown in ***Figure 4—figure supplement 2*** with highlighted bands and sample labels (anti-GFP [for detecting GFP-Ime1], anti-Hxk2).

**Figure supplement 3.** *IME1* transcript levels upon C*LN2* overexpression in *pCUP1-GFP-IME1* background.

**Figure supplement 4.** Ime1 protein levels upon C*LN2* overexpression in *pCUP1-GFP-IME1* background.

**Figure supplement 4—source data 1.** Original file for the immunoblot shown in ***Figure 4—figure supplement 4*** (anti-GFP [for detecting GFP-Ime1], anti-Hxk2).

**Figure supplement 4—source data 2.** Original file for the immunoblot shown in ***Figure 4—figure supplement 4*** with highlighted bands and sample labels (anti-GFP [for detecting GFP-Ime1], anti-Hxk2).

mutant relative to wild type (***Figure 4—figure supplement 1***, ***Figure 4—figure supplement 2***). These data raise the possibility that the meiotic entry defect observed in the *pATG8-CLN2* mutant arises from downregulation of *IME1* expression.

To test whether the increased levels of Ime1 can rescue the meiotic defect of *pATG8-CLN2* mutant, we replaced the endogenous *IME1* promoter with a copper-inducible *CUP1* promoter (***Figure 4D***). We adapted a previously well-characterized overexpression allele of *IME1*, *pCUP1-IME1*, (***Berchowitz et al., 2013***; ***Chia and van Werven, 2016***) and included a functional, N-terminal GFP tag to track Ime1's subcellular localization (*pCUP1-GFP-IME1*). Use of the *CUP1* promoter was successful in elevating *IME1* transcript and protein levels in the presence of *CLN2* overexpression (***Figure 4—figure supplement 3***, ***Figure 4—figure supplement 4***). Despite the rescue of *IME1* expression, gamete formation was still severely perturbed (***Figure 4A***). Using a single z-slice to measure mean nuclear intensity, we noticed that the intensity of the nuclear Ime1 signal was significantly lower in the *pATG8-CLN2; pCUP1-GFP-IME1* cells compared to the *pCUP1-GFP-IME1* control (***Figure 4E and F***, p<0.0001, Mann-Whitney test). This finding indicates that rescue of *IME1* expression did not also rescue sporulation, further suggesting a defect in Ime1 nuclear localization. To test this possibility, we utilized a nanobody trap strategy (***Fridy et al., 2014***) where we C-terminally fused a single-domain anti-GFP antibody to Pus1, a constitutively nuclear localized protein (*PUS1-αGFP*, ***Figure 4G***). In this background, control strains carrying a *GFP-IME1* allele sporulated efficiently, demonstrating that tethering of Ime1 to Pus1 does not interfere with Ime1 function (***Figure 4A***). Furthermore, mean

nuclear intensity of GFP-Ime1 was indistinguishable between *PUS1-αGFP* and *pATG8-CLN2; PUS1-αGFP*, indicating that nuclear localization was fully rescued (*Figure 4H and I*, P=0.6563 [*PUS1-αGFP* vs. *pATG8-CLN2; PUS1-αGFP*], Mann-Whitney test). Surprisingly, these cells still failed to undergo meiosis (*Figure 4A*), suggesting that G1 cyclins interfere with Ime1 function at an additional step beyond misregulating its expression and localization. Alternatively, G1 cyclins could disrupt a different meiotic factor in addition to Ime1.

To induce early meiotic genes, Ime1 must interact with another transcription factor called Ume6 (*Rubin-Bejerano et al., 1996*). Since Ime1 itself does not possess a DNA-binding domain, its binding to Ume6 is essential for targeting Ime1 to early meiotic gene promoters (*Smith et al., 1993*; *Rubin-Bejerano et al., 1996*). To address whether the G1 cyclins might disrupt the interaction between Ime1 and Ume6, we fused the anti-GFP nanobody to Ume6 (*UME6-αGFP*) in the *pATG8-CLN2* strain carrying a GFP tagged Ime1 (*Figure 4J*). This nanobody trap should lead to constitutive tethering of Ime1 to Ume6, as evidenced by the rescue of Ime1 nuclear localization (*Figure 4K and L*, p=0.035 [wild type vs. *pATG8-CLN2 UME6-αGFP*] Mann-Whitney test). Under these conditions, the sporulation defect of *pATG8-CLN2* mutant was rescued, reaching similar levels to wild type (*Figure 4A*). Since *IME1* is expressed from its endogenous promoter in the *pATG8-CLN2; UME6-αGFP* strain, these data suggest that overexpression of G1 cyclins results in meiotic failure due to reduced Ime1-Ume6 interaction.

Our findings thus far highlight the biological significance of restricting SBF activity during the transition from mitotic to meiotic cell fate. Among the SBF targets, G1 cyclins pose a major block to meiotic entry by interfering with Ime1 function, critically at the level of Ime1-Ume6 interaction. While our findings emphasize the importance of *SWI4* regulation to ensure timely SBF activity, the question remains as to how SBF activity is downregulated during transition from mitosis to meiosis.

## Ime1-dependent expression of a LUTI from the *SWI4* locus leads to a reduction in Swi4 protein levels during meiotic entry

Given the importance of *SWI4* downregulation in restricting SBF activity during mitotic to meiotic cell fate transition, we next investigated the mechanism of *SWI4* downregulation. A previous study identified a long undecoded transcript isoform (LUTI) expressed from the *SWI4* locus in meiotic cells (*Brar et al., 2012*, *Figure 5A*). When *SWI4^LUTI* is expressed, the canonical protein-coding *SWI4* transcript, *SWI4^canon*, is downregulated (*Tresenrider et al., 2021*, *Figure 5B*), suggesting that LUTI expression restricts Swi4 protein levels and thus SBF activity during meiotic entry.

To further investigate LUTI-based repression of *SWI4*, we examined the relationship between *SWI4^LUTI* and *SWI4^canon* transcripts using single molecule RNA fluorescence in situ hybridization (smFISH: *Chen et al., 2017*; *Chen et al., 2018*; *Raj et al., 2008*). Two distinct probes were used: one, conjugated to Quasar 670 (Q670) and complementary to the *SWI4* coding sequence (CDS) and the other, conjugated to CAL Fluor Red 590 (CF590) and unique to the 5′ extended LUTI sequence. Accordingly, a spot where the two probe sets colocalized indicated a *SWI4^LUTI* transcript, whereas a spot marked with Q670 probe alone highlighted a *SWI4^canon* transcript. We used a well-established meiotic cell synchronization system (*Berchowitz et al., 2013*) to investigate the precise temporal expression of these two mRNA isoforms. In comparison to premeiotic state, meiotic cells displayed a significant increase in *SWI4^LUTI* transcripts (p<0.0001, Mann-Whitney test) as well as a significant decrease in *SWI4^canon* transcripts (p=0.0007, Mann-Whitney test) (*Figure 5C and D*), thus confirming their inverse expression pattern.

To characterize the functional contribution of the LUTI to *SWI4* downregulation, we eliminated *SWI4^LUTI* production (*ΔLUTI*) by deleting its promoter (*Tresenrider et al., 2021*) and used RNA blotting to visualize the *SWI4* mRNA isoforms. In wild-type cells, *SWI4^LUTI* was readily detectable after 1 hr in SPO, corresponding to early meiotic entry (*Figure 5E*). During the time points when *SWI4^LUTI* was highly expressed, *SWI4^canon* transcript levels were lower, consistent with the smFISH data. In *ΔLUTI* cells, we observed an increase in the abundance of *SWI4^canon* mRNA (*Figure 5E*). This finding corroborates previous reports, where LUTI expression leads to co-transcriptional silencing of the canonical gene promoter, thereby silencing expression of the protein-coding transcript (*Chen et al., 2017*; *Chia et al., 2017*; *Van Dalfsen et al., 2018*; *Tresenrider et al., 2021*; *Wende et al., 2022*). Finally, by immunoblotting, we observed an increase in Swi4 protein abundance in the *ΔLUTI* mutant compared to wild type (*Figure 5E and F*), further indicating that *SWI4^LUTI* expression downregulates Swi4 protein levels in meiosis.

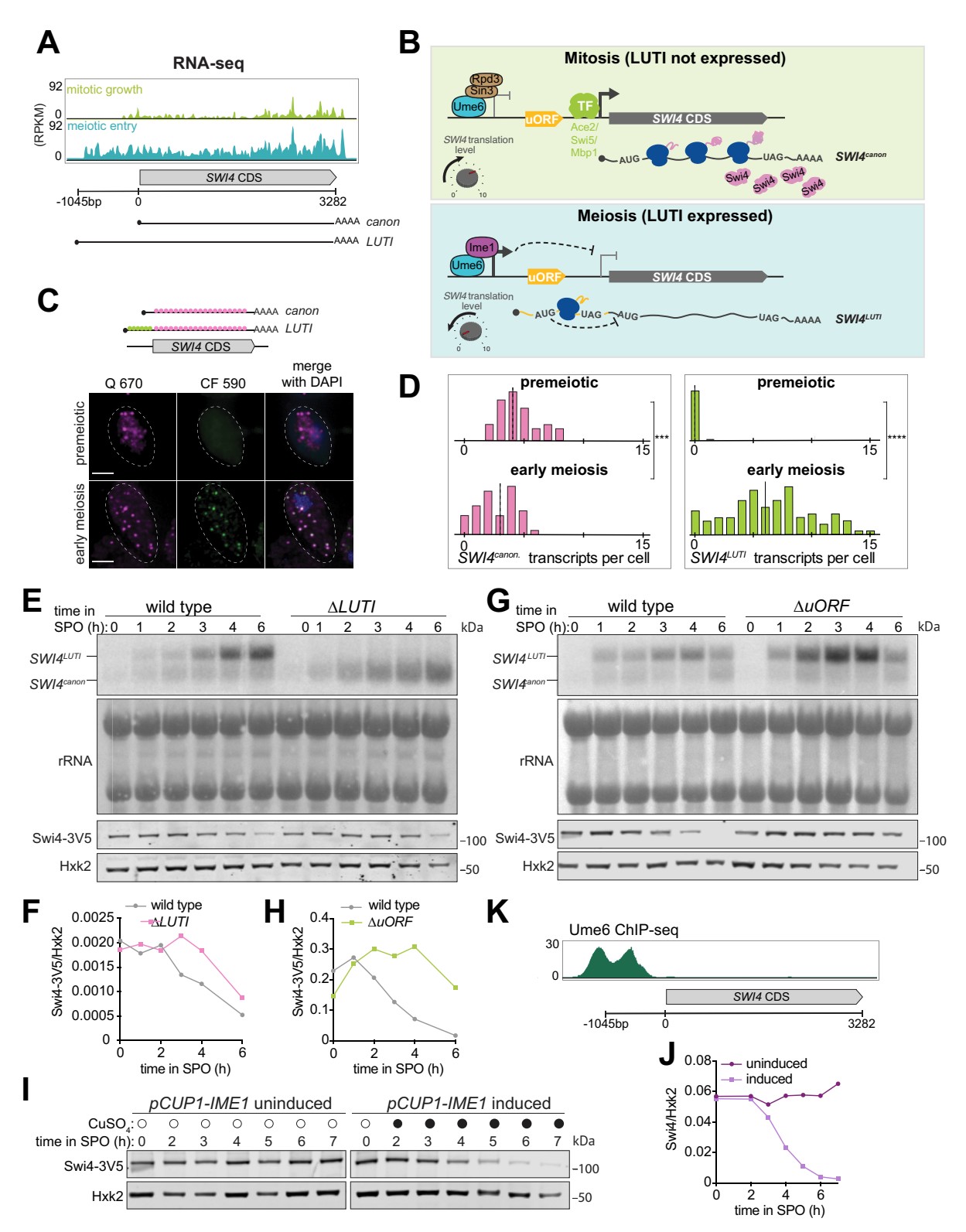

**Figure 5.** Ime1-dependent expression of a LUTI from the *SWI4* locus leads to a reduction in Swi4 protein levels during meiotic entry. (**A**) Genome browser views of RNA-seq data (*Brar et al., 2012*) of the *SWI4* locus. *SWI4^LUTI^* transcription start site is ~1045 bp upstream of *SWI4* ORF translation start site. (**B**) A schematic of LUTI-based gene regulation. Top: Mitotic growth, *SWI4^LUTI^* is repressed due to Ume6-Rpd3-Sin3 complex and *SWI4^canon^* is induced by one or more transcription factors including Ace2, Mbp1, and Swi5, leading to Swi4 protein production. Bottom: Meiosis-specific expression

*Figure 5 continued on next page*

*Figure 5 continued*

of *SWI4^LUTI* by Ime1-Ume6 leads to downregulation of Swi4 protein production due to combined effect of transcriptional and translation interference. *SWI4^LUTI* 5' leader contains 7 AUG uORFs but only one is shown in the model for simplicity. Schematic is adapted from *Tresenrider et al., 2021*. (**C**) Representative smFISH images collected from premeiotic and meiotic cells for detecting *SWI4^canon* and *SWI4^LUTI*. Cells with pCUP1-IME1/pCUP1-IME4 meiotic synchronization system were induced to enter meiosis with 50 µM CuSO₄ after 2 hr in SPO. Premeiotic cells were collected before *IME1/4* induction and meiotic cells were collected 2 hr post *IME1/4* induction from strain UB14273. Q 670 probes (green) hybridize to shared region within *SWI4* CDS. CF590 probes hybridize to the unique 5' leader region of *SWI4^LUTI* (depicted on the schematic shown above the images). DNA was stained with DAPI. Scale bar: 3 µm. (**D**) Quantification of smFISH shown in (**C**), plotted as relative frequency histograms of cells with *SWI4^canon* and *SWI4^LUTI* transcripts per cell. Data pooled from two independent biological replicates. Dashed line indicates median number of transcripts per cell. Each histogram is normalized with maximum bin height being the same across all histograms. A total number of 44 cells counted for premeiotic and 102 cells counted in meiotic prophase. Differences in premeiotic versus meiotic were compared by Mann-Whitney test, two-tailed (***, p=0.0007 [*SWI4^canon*]; ****, p<0.0001 [*SWI4^LUTI*]). (**E**) RNA blot performed on cells collected between 0 and 6 hr in SPO. All strains carry a *SWI4-3V5* tagged allele. Probe was specific for 3V5. Methylene blue staining of rRNA bands was used as a loading control. Matched immunoblotting was performed against Swi4-3V5 using α-V5 and normalized to Hxk2 loading control for each sample. Cells collected are wild type (UB22199) and ΔLUTI (UB23012). Representative blots from one of two biological replicates are shown. (**F**) Quantification of immunoblot in (**E**). (**G**) Performed as described in (**E**) for wild type (UB21386) and ΔuORF (UB23636) strains. Representative blots from one of two biological replicates are shown. (**H**) Quantification of immunoblot in (**G**). (**I**) Immunoblot using α-V5 performed on cells collected between 0 and 7 hr in SPO from a strain carrying pCUP1-IME1 and *SWI4-3V5* alleles (UB34641). Swi4-3V5 abundance was normalized to Hxk2 loading control. Cells were induced to enter meiosis with 50 µM CuSO₄ after 2 hr preincubation in SPO. Representative blots from one of two biological replicates are shown. (**J**) Quantification of immunoblot in (**I**). (**K**) Genome browser view of Ume6-ChIP at the *SWI4* locus (adapted from *Tresenrider et al., 2021*).

The online version of this article includes the following source data and figure supplement(s) for figure 5:

**Source data 1.** Original file for the RNA blot shown in *Figure 5E* (Probe was specific for 3V5, for detecting *SWI4-3V5* canonical and LUTI transcripts).

**Source data 2.** Original file for the RNA blot shown in *Figure 5E* (methylene blue staining for rRNA detection).

**Source data 3.** Original file for the immunoblot shown in *Figure 5E* (anti-V5 [for detecting Swi4-3V5], anti-Hxk2).

**Source data 4.** Original files for the RNA blots and immunoblot shown in *Figure 5E* with highlighted bands and sample labels.

**Source data 5.** Original file for the RNA blot shown in *Figure 5G* (Probe was specific for 3V5, for detecting *SWI4-3V5* canonical and LUTI transcripts).

**Source data 6.** Original file for the RNA blot shown in *Figure 5G* (methylene blue staining for rRNA detection).

**Source data 7.** Original file for the immunoblot shown in *Figure 5G* (anti-V5 [for detecting Swi4-3V5], anti-Hxk2).

**Source data 8.** Original files for the RNA blots and immunoblot shown in *Figure 5G* with highlighted bands and sample labels.

**Source data 9.** Original file for the immunoblot shown in *Figure 5I* (anti-V5 [for detecting Swi4-3V5], anti-Hxk2).

**Source data 10.** Original file for the immunoblot shown in *Figure 5I* with highlighted bands and sample labels (anti-V5 [for detecting Swi4-3V5], anti-Hxk2).

**Figure supplement 1.** Expression of *SWI4^LUTI* in Ume6(T99N) mutant.

*SWI4^LUTI* translation occurs within the upstream ORFs (uORFs) of its 5' leader sequence (*Brar et al., 2012*). If the uORFs repress productive translation of the *SWI4* CDS contained within the LUTI, then their removal should result in increased Swi4 protein levels. To determine whether the uORFs inhibit translation of the *SWI4* CDS, we mutated the start codon of all seven uORFs within *SWI4^LUTI* from ATG to ATC (ΔuORF) and measured Swi4 protein levels using immunoblotting. Compared to wild type, the ΔuORF mutant had higher levels of Swi4 protein (*Figure 5G and H*). RNA blotting confirmed that *SWI4^canon* transcript levels remained similar between wild type and ΔuORF mutant, whereas *SWI4^LUTI* levels were higher in the ΔuORF mutant compared to wild type (*Figure 5G*), likely resulting from increased transcript stability due to bypass of nonsense mediated decay (*Tresenrider et al., 2021*). Our findings indicate that the uORFs within the LUTI are translated at the expense of *SWI4* CDS, thus halting Swi4 synthesis during early meiosis.

Our previous findings highlight an antagonistic relationship between Swi4 and Ime1 nuclear localization, whereby overexpression of *SWI4* leads to a decrease in Ime1 nuclear localization (*Figure 2E*). Given the previous finding that Ime1 activates transcription of LUTIs in early meiosis (*Tresenrider et al., 2021*), we were curious whether the reverse regulation could also occur, where Ime1 causes downregulation of Swi4 through activating *SWI4^LUTI* transcription. To test this possibility, we used an inducible allele of *IME1* (pCUP1-IME1) (*Chia and van Werven, 2016*) and measured Swi4 protein abundance. In the absence of *IME1* induction, Swi4 levels remained constant. However, after 1 hr of *IME1* induction, Swi4 levels started to decrease dramatically (*Figure 5I and J*). Therefore, reduction in Swi4 levels is dependent on Ime1 rather than being driven by nutrient deprivation

in SPO, a condition that is known to trigger autophagy (*Abeliovich and Klionsky, 2001*). To further assess whether *SWI4^LUTI* is regulated by the Ime1-Ume6 transcription factor complex, we analyzed a published Ume6 ChIP-seq dataset (*Tresenrider et al., 2021*) and found evidence for Ume6 binding at the *SWI4^LUTI* promoter (*Figure 5K*). Additionally, analysis of an mRNA-seq dataset from *UME6-T99N* (*Tresenrider et al., 2021*), an allele of *UME6* that can no longer interact with Ime1 revealed a dramatic reduction in *SWI4^LUTI* expression in meiotic conditions (*Figure 5—figure supplement 1*). Together, these data support the notion that Ime1-Ume6 complex induces the expression of *SWI4^LUTI*, which in turn inhibits Swi4 protein synthesis through the combined act of transcriptional and translational interference.

## *SWI4^LUTI* is integrated into a larger regulatory network to regulate SBF activity during meiotic entry

Since the removal of *SWI4^LUTI* resulted in an increase in Swi4 levels, we next wanted to investigate how the loss of LUTI-based regulation affects meiotic entry. We performed mRNA-seq in *ΔLUTI* mutant or wild-type cells during meiotic entry (2 hr in SPO) and used DESeq2 to identify differentially expressed genes. To our surprise, there was no obvious increase in the expression of many SBF targets (*Figure 6A*). Namely, *CLN1* and *CLN2* were both expressed at similar levels to wild type upon loss of *SWI4^LUTI*. Therefore, it appears that disruption of the LUTI-based regulation alone is not sufficient to reactivate SBF targets.

Since transition from the mitotic to meiotic program is a critical cell fate decision, it is likely that additional players are in place to restrict SBF activity. In this case, even when the LUTI-based regulation fails, SBF would remain largely inactive due to a backup mechanism. In support of this notion and consistent with a previous report (*Argüello-Miranda et al., 2018*), the SBF inhibitor Whi5 was expressed during early meiosis (0 hr in SPO) and localized to the nucleus (*Figure 6—figure supplement 1*). During the mitotic G1 phase, Whi5 associates with promoter-bound SBF, thereby preventing the transcription of SBF target genes (*de Bruin et al., 2004*; *Costanzo et al., 2004*). Shortly before the G1/S transition, phosphorylation of Whi5 via the Cln3/CDK pathway activates SBF by promoting nuclear export of Whi5 (*de Bruin et al., 2004*; *Costanzo et al., 2004*). However, in nutrient-deprived conditions that favor meiosis such as nitrogen limitation, *CLN3* is translationally repressed and any Cln3 protein that is expressed is also unstable (*Parviz and Heideman, 1998*; *Gallego et al., 1997*). Without Cln3/CDK, Whi5 is expected to inhibit SBF in a constitutive manner.

To determine whether Whi5 and *SWI4^LUTI* act in parallel to restrict SBF activity, we removed Whi5 from the nucleus using the anchor-away method, which enables compartment-specific depletion of a target protein via inducible dimerization (*Haruki et al., 2008*). Whi5 was tagged with FRB and ribosomal protein Rpl13a was tagged with FKBP12. Upon addition of rapamycin, Rpl13a-FKBP12 and Whi5-FRB formed a heterodimer, leading to successful nuclear exclusion of Whi5 (*Figure 6B*). Using Whi5 anchor away (*WHI5-AA*) alone or in combination with removal of LUTI-based *SWI4* regulation (*ΔLUTI*), we performed mRNA-seq to measure expression of SBF targets and early meiotic genes. Similar to the *ΔLUTI* single mutant, *WHI5-AA* alone did not significantly change the expression of SBF targets or early meiotic genes relative to wild type (*Figure 6C*). However, loss of both modes of regulation resulted in increased expression of SBF targets as well as a concomitant decrease in early meiotic transcripts compared to wild type (*Figure 6D*, *Supplementary file 3*). As expected, Swi4 levels were similar to wild type in *WHI5-AA* but were elevated in *ΔLUTI* and *ΔLUTI; WHI5-AA* mutants (*Figure 6—figure supplement 2*). Gene set enrichment analysis revealed a significant enrichment of the SBF regulon (NES=1.95, p<0.001) as well as significant disenrichment of the early meiotic genes (NES=−3.39, p<0.001) in the *ΔLUTI; WHI5-AA* double mutant (*Figure 6—figure supplement 3*).

Finally, we monitored meiotic entry using time-lapse fluorescence microscopy in strains carrying endogenously tagged Rec8-GFP in conjunction with Htb1-mCherry. In agreement with our mRNA-seq data, loss of either LUTI- or Whi5-based repression of SBF alone was not sufficient to cause a delay in meiotic entry (*Figure 6E*). However, simultaneous perturbation of both pathways led to a significant delay in meiotic entry (*Figure 6E*, p=0.0112, Mann-Whitney test). These data further support the notion that *SWI4^LUTI* is integrated into a larger regulatory network to regulate SBF activity during meiotic entry, which includes Whi5-mediated repression of SBF.

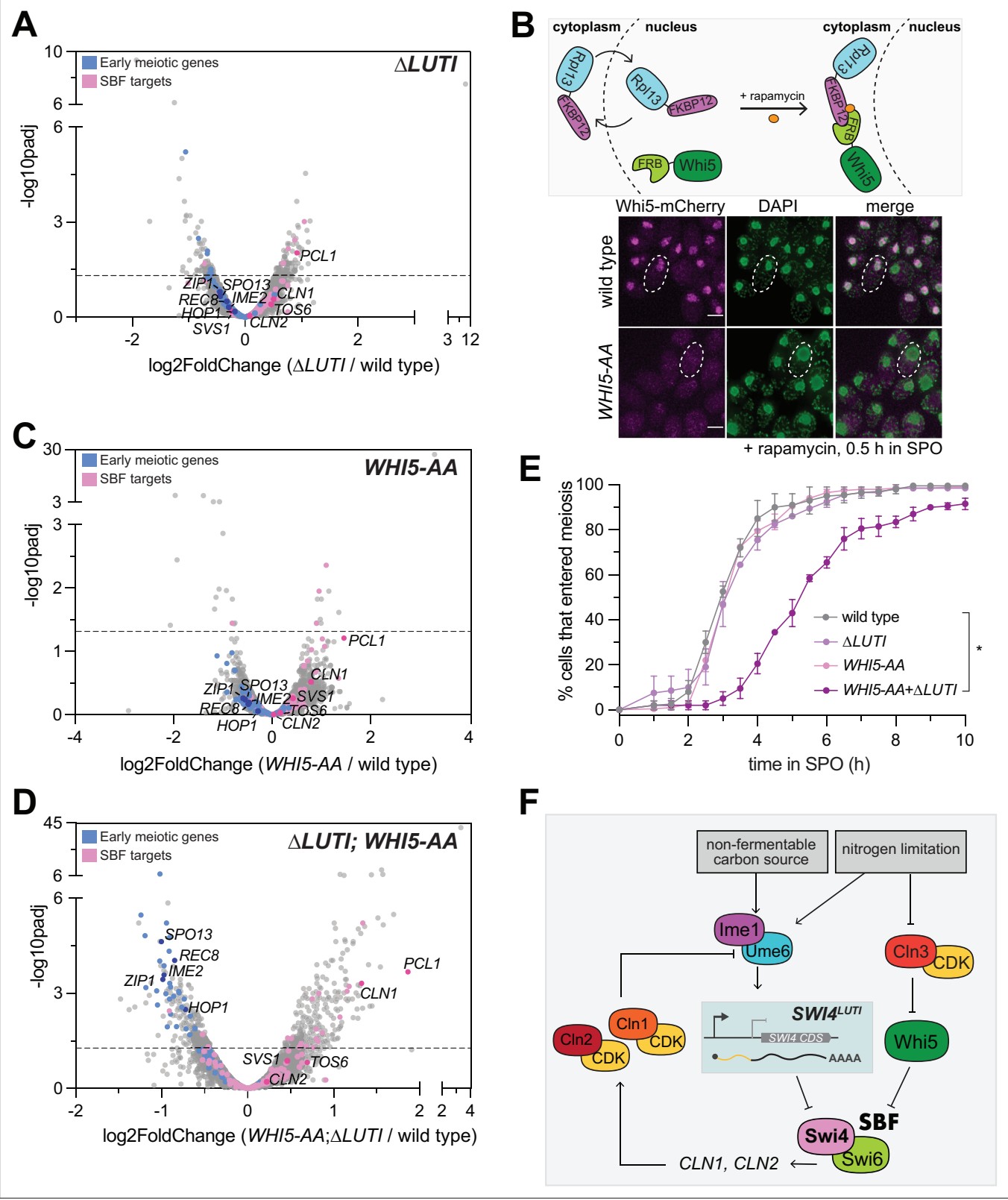

**Figure 6.** *SWI4^LUTI* is integrated into a larger regulatory network to regulate SBF activity during meiotic entry. (**A**) Volcano plot of DESeq2 analysis for *ΔLUTI* versus wild type. Dashed line indicates padj (p-value)=0.05. Analysis was performed with mRNA-seq in duplicate. Wild type (UB27083) and *ΔLUTI* (UB26874) collected at 2 hr in SPO. SBF targets (pink) and early meiotic genes (blue) defined by *Iyer et al., 2001* and; *Brar et al., 2012*. Darker pink or darker blue, labeled dots are well studied targets in either gene set list. (**B**) Top: Schematic of the anchor-away system using *WHI5-mCherry-FRB* (*WHI5-*

*Figure 6 continued*

*AA*) and *RPL13a-FKBP12* alleles. Bottom: Fixed imaging of cells marked with *WHI5-mCherry-FRB* (*WHI5-AA*) with DNA stained with DAPI. One µM rapamycin added at 0 hr in SPO to induce nuclear exclusion of Whi5 (UB25431) strain collected at 0.5 hr in SPO. Scale bar: 3 µm. Cells are rapamycin resistant due to mutated *TOR1* (*tor1-1*) and frp1Δ (yeast FKBP12 homolog) to reduce competition between for binding of Frb and Fkbp12. (**C**) Same as in (**A**) but for wild type (UB27083) and *WHI5-AA* (UB25431) collected at 2 hr in SPO. (**D**) Same as in (**A**) but for wild type (UB27083) and *ΔLUTI; WHI5-AA* (UB25428) collected at 2 hr in SPO. (**E**) Live-cell imaging of cells in meiosis marked by Rec8-GFP and nuclear marker Htb1-mCherry. The following genotypes were imaged: wild type (UB35987), *ΔLUTI* (UB35989), *WHI5-AA* (UB35991), *ΔLUTI; WHI5-AA* (UB35989). Quantification as percent of cells that entered meiosis assayed by nuclear Rec8 appearance. Experiments were performed using two biological replicates, mean value plotted with range. Differences in meiotic progression compared by Mann-Whitney test, two-tailed (*, p=0.0112 [wild type vs. *ΔLUTI; WHI5-AA*]). (**F**) Model of SBF regulation during meiotic entry. Ime1 downregulates Swi4 protein expression via induction of *SWI4^{LUTI}* while Whi5 represses SBF activity in parallel to LUTI-based mechanism to prevent expression of SBF targets, including G1 cyclins, which perturb meiotic entry via blocking interaction between Ime1 and its cofactor Ume6.

The online version of this article includes the following source data and figure supplement(s) for figure 6:

**Figure supplement 1.** Whi5 localization in early meiosis.

**Figure supplement 2.** Swi4 protein levels matching *Figure 6A, C and D*.

**Figure supplement 2—source data 1.** Original file for the immunoblot shown in *Figure 6—figure supplement 2* (WT and *ΔLUTI*; anti-V5 [for detecting Swi4-3V5], anti-Hxk2).

**Figure supplement 2—source data 2.** Original file for the immunoblot shown in *Figure 6—figure supplement 2* (*WHI5-AA* and *WHI5-AA, ΔLUTI*; anti-V5 [for detecting Swi4-3V5], anti-Hxk2).

**Figure supplement 2—source data 3.** Original file for the immunoblot shown in *Figure 6—figure supplement 2* with highlighted bands and sample labels (anti-V5 [for detecting Swi4-3V5], anti-Hxk2).

**Figure supplement 3.** GSEA data for *ΔLUTI; WHI5-AA*.

## Discussion

In this study, we describe two distinct mechanisms by which meiotic cells inhibit the activity of SBF, a transcription factor complex crucial for the G1/S transition during the mitotic cell cycle. The dual inhibition of SBF converges on its unique subunit Swi4 and is important for proper transition from mitotic to meiotic cell fate. Swi4 inhibition occurs through: (1) meiosis-specific expression of a LUTI from the *SWI4* locus, *SWI4^{LUTI}*, which downregulates Swi4 protein levels, and (2) repression of Swi4 activity by Whi5, which occurs even when the LUTI-based regulation is disrupted. Conditions that lead to elevation of Swi4 levels at meiotic entry result in abrupt activation of the SBF and a concomitant decline in the expression of early meiotic genes, leading to meiotic progression delays. Additionally, we show that the SBF targets *CLN1* and *CLN2* are major drivers of the meiotic entry defect resulting from SBF misregulation. The *CLN*-dependent phenotypes can be largely rescued by tethering of the central meiotic regulator Ime1 to its cofactor Ume6, suggesting that the primary reason for meiotic failure caused by these G1 cyclins is due to a defect in Ime1-Ume6 interaction. Our findings reveal the functional role of a LUTI in establishing the meiotic transcriptional program, demonstrate how the LUTI-based regulation is integrated into a larger regulatory network to ensure timely SBF activity, and provide mechanistic insights into how SBF misregulation impedes transition from mitotic to meiotic cell fate (*Figure 6F*).

### Rewiring of the G1/S regulon

It has long been known that entry into the mitotic cell cycle and meiotic differentiation are regulated differently. During the mitotic cell cycle, budding yeast cells must reach a critical size before commitment to division in late G1 phase, an event termed the 'Start' (*Hartwell et al., 1974*; *Johnston et al., 1977*). Start depends on the activation of the G1/S regulon by SBF and MBF, which regulate ~200 genes that function in polarized growth, macromolecular biosynthesis, DNA replication, and repair among other critical processes (reviewed in *Jorgensen and Tyers, 2004*). In contrast, entry into meiosis requires the transcriptional activator, Ime1, which is exclusively induced in diploid cells through the integration of several extrinsic cues including nitrogen levels, carbon source, and extracellular pH (reviewed in *van Werven and Amon, 2011*). Ime1 initiates the first transcriptional wave of early meiotic genes required for DNA replication, recombination, and chromosome morphogenesis. Before our study, it was unclear whether and how SBF-MBF activity is regulated during the transition from mitosis to meiosis, despite the transcriptome-wide differences between these two developmental

programs. We found that during meiotic entry, the SBF-specific Swi4 subunit is downregulated, which indicates that the activity of SBF, but not MBF, is constrained. Consistently, SBF-specific targets display either low or no expression during early meiosis, while MBF targets are upregulated upon meiotic entry. MBF targets are involved in processes shared between mitosis and meiosis including DNA replication and repair, whereas SBF targets are predominantly involved in processes unique to mitosis such as budding and cell wall biosynthesis. This functional specialization might have enabled adaptations to accommodate specific cell fates. Likewise, SBF-specific targets that are detrimental to meiosis, such as the G1 cyclins (*Colomina et al., 1999*, and this study), might contribute to the mutually exclusive nature of the mitotic and meiotic programs.

Absolute measurements of the concentrations of SBF and MBF in single cells during mitotic G1 demonstrated that these transcription factors are sub-saturating with respect to their target promoters in small cells, but that SBF and MBF levels increase as cells grow, suggesting that their abundance is a limiting factor in activating the G1/S regulon (*Dorsey et al., 2018*). Corroborating these findings, we show that overexpression of *SWI4* as well as the simultaneous disruption of the two pathways that normally restrict SBF activity at meiotic entry – namely LUTI and Whi5 – result in abrupt activation of the SBF regulon and a concomitant decrease in early meiotic gene expression. However, MBF targets do not seem to be affected in response to increased SBF activity. This is surprising given that SBF and MBF share a common subunit, Swi6, and that an upsurge in Swi4 abundance might be expected to titrate Swi6 away from MBF. It is possible that meiosis-specific regulators like Ime1 and Ime2 compensate to maintain MBF target expression (*Brush et al., 2012*). Future studies are necessary to dissect the meiotic roles of MBF and potential compensatory mechanisms.

## Inhibition of G1 cyclins during meiotic entry and vice versa

We found that the meiotic entry delay due to untimely SBF activation can be partially rescued by loss of either *CLN1* or *CLN2*, demonstrating that both cyclins are responsible for the meiotic defects associated with SBF misregulation. In addition to *CLN1* and *CLN2*, improper activation of SBF also leads to upregulation of *PCL1* during meiotic entry (*Figures 2F and 6B*). *PCL1* encodes a cyclin that interacts with the Pho85 CDK and is involved in the regulation of polarized cell growth and morphogenesis as well as progression through the cell cycle (*Espinoza et al., 1994*). Whether *PCL1* misexpression contributes to the SBF-associated meiotic entry defects is an interesting avenue for future studies.

We and others have demonstrated that elevated levels of G1 cyclins inhibit meiotic entry. Therefore, it is crucial to dissect the mechanisms that govern the downregulation of G1 cyclins during this process. Among the G1 cyclins that are known to inhibit meiotic entry (*Colomina et al., 1999*), only the mechanism of *CLN3* restriction was previously known (*Parviz and Heideman, 1998*; *Gallego et al., 1997*). Our study provides a mechanistic understanding of how *CLN1* and *CLN2* remain quiescent during meiotic entry through the combined act of LUTI-based and Whi5-dependent inhibition of SBF. Whether additional mechanisms play a role in restricting SBF and/or *CLN1-CLN2* remains to be determined.

Our findings also shed light on the mechanism by which G1 cyclins prevent meiotic entry. Previous work demonstrated that G1 cyclin overexpression leads to downregulation of *IME1* expression and inhibition of Ime1 nuclear localization (*Colomina et al., 1999*). Even though we observed similar defects in *IME1* in response to *CLN2* overexpression, increasing *IME1* expression or targeting Ime1 to the nucleus did not result in successful meiosis (*Figure 4*). Instead, we found that restoring the interaction between Ime1 and its cofactor Ume6 was sufficient to bypass the meiotic blockage exerted by Cln/CDK. Collectively, our analyses suggest that the primary reason why G1 cyclins cause a meiotic failure is due to a deficiency in Ime1-Ume6 interaction. A previous study reported that G1 cyclins do not affect the interaction between Ime1 and Ume6 (*Colomina et al., 1999*). However, this conclusion was based on the use of a truncated Ime1 protein that only contained the Ume6 interaction domain. Therefore, CLN-dependent regulations on full-length Ime1 might be missed in this context.

Future work could be aimed at dissecting how G1 cyclins affect the interaction between Ime1 and Ume6 and whether their impact on Ime1's subcellular localization is primarily due to G1 cyclin-dependent changes in Ime1-Ume6 interaction. Since there are no CDK consensus phosphorylation sites on Ime1 itself, other players are likely to be involved. Rim11 and Rim15 are potential candidates since these two kinases have been implicated in Ime1-Ume6 phosphorylation as well as regulation of Ime1 localization and its interaction with Ume6 (*Rubin-Bejerano et al., 1996*; *Vidan and Mitchell,*

1997; *Bowdish et al., 1994*). Interestingly, Rim15 contains CDK consensus phosphorylation sites (*Holt et al., 2009*; *Moreno-Torres et al., 2017*; *Breitkreutz et al., 2010*), Furthermore, Cln2/CDK activity has been previously shown to inhibit Rim15 nuclear localization (*Talarek et al., 2010*), thereby making it an attractive candidate for further investigation.

## Antagonistic relationship between two key transcription factors, Swi4 and Ime1

Swi4 and Ime1 regulate each other in an antagonistic manner (*Figure 6F*), as supported by several observations. First, overexpression of *SWI4* leads to a reduction in Ime1 nuclear localization (*Figure 2B*). Second, premature activation of SBF through *SWI4* overexpression or simultaneous disruption of *SWI4^LUTI^* and Whi5 pathways results in a significant delay in the chromosomal localization of Rec8, a direct transcriptional target of Ime1 (*Figures 2H, I and 6E*). Additionally, many early meiotic genes that are also transcriptional targets of Ime1 are downregulated under the same conditions (*Figures 2F and 6D*). Third, *IME1* is inhibited by the G1 cyclins, which are themselves targets of Swi4/SBF (*Figure 4*). The reverse regulation, where Ime1 inhibits Swi4, is also in place. This occurs through the Ime1-dependent expression of *SWI4 ^LUTI^*, which is necessary to downregulate Swi4 levels (*Figure 5*, also reported in *Tresenrider et al., 2021*). The antagonistic relationship between Swi4 and Ime1 is further evidenced by their mutually exclusive pattern of nuclear localization at the single-cell level (*Figure 2E*). In summary, our study highlights the multiple ways in which Swi4 and Ime1 regulate each other, which likely plays a crucial role in cellular decision making between the mitotic and meiotic transcriptional programs.

## Intersection of LUTI-based gene repression with other regulatory pathways

While LUTI-based regulation is both necessary and sufficient to downregulate Swi4 levels, disrupting it alone is not enough to activate the SBF regulon. We found that the LUTI-based mechanism works together with the Whi5 pathway to inhibit SBF activity (*Figure 6*). As a result, only when both mechanisms are simultaneously disrupted, do cells exhibit abrupt activation of SBF targets and subsequent meiotic entry defects. This two-pronged inhibition of SBF activity is reminiscent of how meiotic cells prevent microtubule-kinetochore interactions during prophase I (*Miller et al., 2012*). Specifically, LUTI-based regulation represses the expression of a limiting outer kinetochore subunit, Ndc80 (*Chen et al., 2017*). Even when the outer kinetochore assembles upon disruption of *NDC80^LUTI^*, microtubules are still unable to engage with the kinetochores due to restriction of Clb/CDK activity. The loss of both regulations results in meiotic chromosome segregation defects (*Chen et al., 2017*). Integrating LUTI-based repression into larger regulatory networks likely ensures robustness in cellular decision-making, providing a fail-safe system. However, combinatorial use of the LUTI-based mechanism with other regulatory pathways poses a challenge to uncover their functional importance. Despite 8% of the yeast genes being subject to LUTI-based regulation during meiosis (*Cheng et al., 2018*), the biological significance of only two LUTIs, *NDC80^LUTI^* and *SWI4^LUTI^*, have been uncovered thus far. Therefore, systematic studies aiming to dissect the biological roles of LUTIs in meiosis and beyond would benefit from simultaneous perturbation approaches, including synthetic genetic interactions.

## Concluding remarks

Cell state changes are primarily governed by transcription factors. Our study highlights how transition from mitotic to meiotic cell fate is ensured by inhibition of the SBF transcription factor through two distinct mechanisms. The SBF regulon includes G1 cyclins, whose untimely expression blocks meiotic entry by disrupting the interaction between Ime1 and Ume6, a key transcriptional co-activator of meiosis. SBF and Ime1-Ume6 antagonize each other, thereby helping establish a mutually exclusive state between the mitotic and meiotic transcriptional programs. Mammalian cells have functional homologs to both sets of transcription factors. First, similar to Ume6, MEIOSIN binds to the promoters of early meiotic genes and recruits Ime1-like STRA8 to activate gene expression (*Anderson et al., 2008*; *Kojima et al., 2019*; *Baltus et al., 2006*; *Ishiguro et al., 2020*). Second, SBF and MBF counterparts in mammalian cells are known as E2Fs, nine of which control the G1/S transition (reviewed in *van den Heuvel and Dyson, 2008*). Each E2F regulates a distinct sets of genes (*Gaubatz et al., 2000*). Furthermore, some members of the E2F family are involved in tissue-specific regulation of cell fate

(*Julian et al., 2016*). However, the molecular mechanisms driving the rewiring of these complex gene networks are not well understood. Our investigation could help shed light on the regulatory mechanisms governing the interplay between mitotic and meiotic transcription factors outside of yeast.

# Materials and methods

## Key resources table

| Reagent type (species) or resource | Designation | Source or reference | Identifiers | Additional information |
|---|---|---|---|---|
| Strain, strain background (*Saccharomyces cerevisiae*) | wild type | Amon lab | SK1 | see **Supplementary file 4** for strain genotypes |
| Antibody | anti-V5 (Mouse monoclonal) | ThermoFisher Scientific | R960-25, RRID:AB_2556564 | 1:2000 |
| Antibody | anti-GFP (Mouse monoclonal) | Takara | 632381, RRID:AB_2313808 | 1:2000 |
| Antibody | anti-Hxk2 (Rabbit monoclonal) | US Biological | H2035-01, RRID:AB_2629457 | 1:20000 |
| Antibody | anti-mouse conjugated to IRDye 800CW (Mouse monoclonal) | LI-COR Biosciences | 926–32212, RRID:AB_621847 | 1:20000 |
| Antibody | anti-rabbit conjugated to IRDye 680CW (Rabbit monoclonal) | LI-COR Biosciences | 926–68071, RRID:AB_10956166 | 1:20000 |
| Antibody | anti-rabbit conjugated to IRDye 800CW (Rabbit monoclonal) | LI-COR Biosciences | 926–68073, RRID:AB_10954442 | 1:20000 |
| Antibody | anti-Swi4 (Rabbit polyclonal) | **Andrews and Herskowitz, 1989** | | 1:2000 |
| Antibody | anti-Swi6 (Rabbit polyclonal) | **Harris et al., 2013** | | 1:2000 |
| Antibody | anti-Mbp1 (Rabbit polyclonal) | **Harris et al., 2013** | | 1:2000 |
| Recombinant DNA reagent | pUB595_pFA6a-FRB-KanMX6 | **Haruki et al., 2008** | | |
| Recombinant DNA reagent | pUB1585_LEU2-pATG8-SWI4-linker-3V5 | This paper | | contact Ünal lab to obtain plasmid |
| Recombinant DNA reagent | pUB1587_LEU2-pSWI4(−1200—1)-SWI4-3V5-3'UTR | This paper | | contact Ünal lab to obtain plasmid |
| Recombinant DNA reagent | pUB1588_LEU2-pSWI4(−1200 to −934) Δ-SWI4-3V5-3'UTR (LUTIΔ) | This paper | | contact Ünal lab to obtain plasmid |
| Recombinant DNA reagent | pUB1734_LEU2-pSWI4(ATG >ATC mutant)-SWI4-3V5-3'UTR (uORFΔ) | This paper | | contact Ünal lab to obtain plasmid |
| Recombinant DNA reagent | pUB1899_HIS3-pATG8-CLN2-linker-3V5 | This paper | | contact Ünal lab to obtain plasmid |
| Recombinant DNA reagent | pUB2144_TRP1-pATG8-CLN1-linker-3V5 | This paper | | contact Ünal lab to obtain plasmid |
| Sequence-based reagent | 6852_CLN2_F | This paper | | TCGTGTTACGGGACCAAGCC |
| Sequence-based reagent | 6853_CLN2_R | This paper | | TACGTGCCCTTGGGTTGGGA |
| Sequence-based reagent | 6887_CLN1_F | This paper | | ACGTCTCCATCCCCACAGGT |
| Sequence-based reagent | 6888_CLN1_R | This paper | | CGGACCCGCCGCAATAATGA |
| Sequence-based reagent | 3301_PFY1_F | This paper | | ACGGTAGACATGATGCTGAGG |
| Sequence-based reagent | 3302_PFY1_R | This paper | | ACGGTTGGTGGATAATGAGC |
| Sequence-based reagent | 2081_IME1_F | This paper | | TCACCACCGCCATCACTACA |
| Sequence-based reagent | 2082_IME1_R | This paper | | TGAAGGAGTAAGCCGCAGCA |
| Sequence-based reagent | 6854_CDC21_F | This paper | | TTGGCCGGTGATACAGACGC |
| Sequence-based reagent | 6855_CDC21_R | This paper | | ACGGGCCCCAGATCTCCTAC |
| Sequence-based reagent | 6858_RNR1_F | This paper | | ACCCTAGCGGCCAGAATTGC |
| Sequence-based reagent | 6859_RNR1_R | This paper | | CATGGGAGCGGGCTTACCAG |
| Sequence-based reagent | 2598_ACT1_F | This paper | | GTACCACCATGTTCCCAGGTATT |
| Sequence-based reagent | 2599_ACT1_R | This paper | | AGATGGACCACTTTCGTCGT |

*Continued on next page*

*Continued*

| Reagent type (species) or resource | Designation | Source or reference | Identifiers | Additional information |
|---|---|---|---|---|
| Sequence-based reagent | 5429_SWI4LUTI_F | This paper | | ACAAGGACTAAGAAGCACGTCA |
| Sequence-based reagent | 5430_SWI4LUTI_R | This paper | | ACCAATGCTAAAGGATGGCA |
| Sequence-based reagent | 5918_3 V5_probe_F | *Tresenrider et al., 2021* | | CTAGTGGATCCAGGTAAACCTAT |
| Sequence-based reagent | 2921_3 V5_probe_R | *Tresenrider et al., 2021* | | TAATACGACTCACTATAGGCCAGTCCTAATAGAGGATTAGG |
| Commercial assay, kit | NEXTflexTM Rapid Directional mRNA-Seq Kit | Perkin Elmer | NOVA-5138 | |
| Commercial assay, kit | Prime-It II Random Primer Labeling Kit | Agilent Technologies, Inc | 300385 | |
| Commercial assay, kit | MinElute PCR Purification Kit | QIAGEN | 28004 | |
| Commercial assay, kit | MAXIscript T7 Transcription Kit | Thermo Fisher Scientific | AM1312 | |
| Commercial assay, kit | TURBO DNA-free Kit | Thermo Fisher Scientific | AM1907 | |
| Commercial assay, kit | Superscript III kit | Thermo Fisher Scientific | 18080044 | |
| Commercial assay, kit | HiFi DNA Assembly Master Mix | New England Biolabs | E2621 | |
| Commercial assay, kit | Absolute Blue qPCR Mix | ThermoFisher Scientific | AB4162B | |
| Software, algorithm | FIJI | *Schindelin et al., 2012* | | |
| Software, algorithm | softWoRx, 6.5.2 | Cytiva | | |
| Software, algorithm | Hisat2 | *Kim et al., 2019* | | |
| Software, algorithm | StringTie | *Pertea et al., 2015* | | |
| Software, algorithm | DESeq2, v1.34.0 | *Love et al., 2014* | | |
| Software, algorithm | Image Studio Lite | LI-COR | RRID:SCR_013715 | |
| Software, algorithm | GraphPad Prism | GraphPad Software | RRID:SCR_002798 | |
| Software, algorithm | GSEA, v4.3.2 | *Subramanian et al., 2005* | | |
| Software, algorithm | Go Slim Mapper | SGD | | |
| Software, algorithm | smFISH quantification | *Chen et al., 2017* | | |
| Other | Semiwet Transfer Buffer | Bio-Rad | 10026938 | transfer buffer for immunoblot |
| Other | Intercept (PBS) Blocking Buffer | LI-COR Biosciences | 927–70001 | blocking buffer for immunoblot |
| Other | ULTRAhyb Ultrasensitive Hybridization Buffer | Thermo Fisher Scientific | AM8669 | buffer for northern blot |
| Other | Amersham Hybond-N+ | Cytiva | RPN203B | membrane for northern blot |
| Other | NucAway Spin Columns | ThermoFisher Scientific | AM10070 | columns for northern blot |
| Other | AMPure XP beads | Beckman Coulter | A63881 | beads for library prep. |
| Other | High Sensitivity D1000 Reagents | Agilent | 5067–5585 | tape station reagent |
| Other | High Sensitivity D1000 ScreenTape | Agilent | 5067–5584 | tape station reagent |
| Other | DAPI | Sigma | D9564 | fluorescence microscopy |
| Other | Concanavalin A | Sigma | C7642 | fluorescence microscopy |

## Strain construction and cloning

All strains used in this study were derived from the SK1 background. Detailed information about the strain genotypes can be found in *Supplementary file 4*.

For *SWI4* (pUB1587) 1200 bases upstream of the ORF, *SWI4* ORF, C-terminal 3V5 epitope tag, and *SWI4* 3'UTR (1000 bases downstream of ORF) were cloned into a *LEU2* single integration vector by Gibson assembly (NEB) (*Gibson et al., 2009*). LUTI promoter deletion (Δ*LUTI*) strain (pUB1588) was similarly constructed with −1200 to −934 bases upstream from ORF deleted. For the uORF mutant (Δ*uORF*) strain (pUB1734) all seven ATG uORFs were mutated using gBlocks with the ATG >ATC mutations and cloned by Gibson assembly into 3V5-tagged *LEU2* single integration vector. In all strains described above, the endogenous *SWI4* gene is deleted using Pringle-based insertion of KanMX6 marker (*Bähler et al., 1998*).

For overexpression of *SWI4* (*pUB1585*), *CLN1* (*pUB2144*), and *CLN2* (*pUB1899*), the relevant gene carrying a 3V5 epitope tag was cloned downstream of the *ATG8* promoter, which is highly expressed during meiosis (*Brar et al., 2012*). A *LEU2* single integration vector was used for cloning the fragments by Gibson assembly (*NEB*). In the strains carrying overexpression transgenes, the wild-type alleles at the endogenous loci remained intact. *pCUP1-GFP-IME1* allele was made with Pringle-based insertion of the *pCUP1* promoter upstream of *GFP-IME1*. PUS1-αGFP and UME6-αGFP were made by Pringle-based insertion using a plasmid (pUB1707) gifted by the Lackner Lab. For *WHI5-AA* the *WHI5-mCherry-FRB* allele was made by Pringle-based insertion using a plasmid (pUB595) gifted by the Nasmyth lab. All plasmid sequences were confirmed by sequencing.

All single integration plasmids were digested with Pme1 before transformation. Proper integration was confirmed by PCR. The plasmids constructed in this study are listed in *Supplementary file 5*.

## Sporulation conditions

For meiotic experiments using *pCUP1-IME1/pCUP1-IME4* or *pCUP-IME1* system, cells were synchronized as described in *Chia and van Werven, 2016*. Briefly, after 24 hr of growth in YPD at room temperature, saturated cultures (OD$_{600}$ >10) were diluted to an OD$_{600}$ of 0.25 in BYTA (1% yeast extract, 2% bacto tryptone, 1% potassium acetate, and 1.02% potassium phthalate) for 16–18 hr of growth at 30 °C (OD$_{600}$ of >5). Cells were washed with water twice before final resuspension in SPO with 0.5% potassium acetate to an OD$_{600}$ of 1.85. After 2 hr in SPO, *IME1* and *IME4* were induced with 50 µM CuSO$_4$. Sporulation efficiency was measured after 24 hr in SPO. Anchor away meiotic experiments was performed as described above with final 1 µM rapamycin (Millipore) added to BYTA 30 min before transfer to SPO and again 1 µM rapamycin added to SPO media.

In all other meiotic experiments, cells were prepared as in *Carlile and Amon, 2008*. Briefly, after 24 hr of growth in YPD at room temperature, saturated cultures (OD$_{600}$ >10) were diluted to an OD$_{600}$ of 0.25 and inoculated in BYTA (1% yeast extract, 2% bacto tryptone, 1% potassium acetate, and 1.02% potassium phthalate) for 16–18 hr at 30 °C (OD$_{600}$ of >5). Cells were washed with water twice before final resuspension in SPO with 0.5% potassium acetate to an OD$_{600}$ of 1.85. Sporulation efficiency was counted after 24 hr in SPO.

*UME6-αGFP* meiotic experiments were prepared as in *Chia and van Werven, 2016*. Briefly, after 18 hr of growth in YPD at room temperature, saturated cultures (OD$_{600}$ >10) were diluted to an OD$_{600}$ of 0.2 in reduced YPD (1% yeast extract, 2% peptone, 2% dextrose, uracil [24 mg/L]). Reduced YPD was used instead of BYTA to prevent cells from prematurely entering meiosis in BYTA due to the Ime1-Ume6 interaction from the GFP nanobody. Cells were grown for ~6 hr at 30 °C until they reached an OD$_{600}$ between 0.5 and 1.0. Cultures were then back diluted to OD$_{600}$ of 0.1 and grown for 18 hr at 30 °C. Cells were washed with water twice before final resuspension in SPO with 0.5% potassium acetate to an OD$_{600}$ of 1.85. Sporulation efficiency was counted after 24 hr in SPO.

## RNA extraction for mRNA-seq, RT-qPCR, and RNA blotting

RNA extraction was performed as described in *Tresenrider et al., 2021*. Briefly, ~4 OD unit of cells were pelleted by centrifugation for 1 min at 20,000 rcf and snap frozen in liquid nitrogen. Cells were thawed on ice and resuspended in TES buffer (10 mM Tris pH 7.5, 10 mM EDTA, 0.5% SDS). An equal volume of Acid Phenol:Chloroform:Isoamyl alcohol (125:24:1; pH 4.7) was added to cells and incubated at 65 °C for 45 min shaking at 1400 rpm. Aqueous phase was transferred to a new tube with chloroform, vortexed for 30 s, separated by centrifugation, and precipitated in isopropanol and sodium acetate overnight at −20 °C. Pellets were washed with 80% ethanol and resuspended in DEPC water for 10 min at 37 °C. Total RNA was quantified using a nanodrop.

## mRNA sequencing (mRNA-seq) and analysis

RNA-seq libraries were generated with the NEXTflex Rapid Directional mRNA-Seq Kit (NOVA-5138, Perkin Elmer). 10 µg of total RNA was used as input for all libraries. AMPure XP beads (A63881, Beckman Coulter) were used to select fragments between 200 and 500 bp. Libraries were quantified using the Agilent 4200 TapeStation (Agilent Technologies, Inc). Samples were submitted for 100 bp SE sequencing by the Vincent J. Coates Genomics Sequencing Laboratory with a NovaSeq SP 100 SR.

Hisat2 (*Kim et al., 2019*) was used to align reads to map sequences to SK1 PacBio genome. Quantification of RNA as transcripts per million was done StringTie (*Pertea et al., 2015*). Fold-change quantification was performed by DESeq2 using default options (version 1.34.0, *Love et al., 2014*).

For Gene Set Enrichment Analysis (GSEA) v4.3.2 was used to compare TPM values for different gene sets. The early meiotic gene set was created from Figure 2 of *Brar et al., 2012*. The SBF regulon was from Figure 3 of *Iyer et al., 2001*. GSEA was performed on the desktop app with default settings expect 'Collapse/Remap to gene symbols' was set to 'No_Collapse' and 'Permutation type' was set to 'gene_set'.

For Gene Ontology (GO) Analysis, SGD Gene Ontology Slim Term Mapper was used for GO analysis using Yeast Go-Slim: process GO Set. Used output from DE-Seq2 analysis with a cutoff of padj (p value)<0.05.

The two overlapping targets (*SWE1* and *TOS4*) between the early meiotic gene set and SBF targets are not plotted on volcano plot or used for GSEA analysis.

## Reverse transcription-quantitative polymerase chain reaction (RT-qPCR)

Five µg of isolated total RNA was treated with DNase (TURBO DNA-free Kit). cDNA was reverse transcribed following the Superscript III kit (Thermo Fisher Scientific). Quantification was performed with Absolute Blue qPCR Mix (Thermo Fisher Scientific). Meiotic samples were normalized to *PFY1*, and mitotic samples were normalized to *ACT1*. Oligonucleotides are listed in *Supplementary file 6*.

## Protein extraction and immunoblotting

Approximately 4 OD$_{600}$ of cells were collected and pelleted by centrifugation for 1 min at 20,000 rcf. Pellet was resuspended in 5% (w/v) TCA for at least 15 min at 4 °C. Cells were washed with TE50 (50 mM Tris-HCl [pH 7.5], 1 mM EDTA) and then with 100% acetone. The cell pellet was dried overnight and then lysed with glass beads in lysis buffer (Tris-HCl [pH 7.5]), 1 mM EDTA, 2.75 mM DTT, protease inhibitor cocktail (cOmplete EDTA-free [Roche]). Next 3 x SDS sample buffer (187.5 mM Tris [pH 6.8], 6% β-mercaptoethanol, 30% glycerol, 9% SDS, 0.05% bromophenol blue) was added and the cell lysate was boiled for 5 min at 95 °C. Protein was separated by PAGE using 4–12% Bis-Tris Bolt gels (Thermo Fisher) and transferred onto 0.45 µm nitrocellulose membranes.

Cln1-3V5, Cln2-3V5 and GFP-GFP tagged proteins were all transferred onto 0.45 µm nitrocellulose membranes using a semi-dry transfer apparatus (Trans-Blot Turbo System (Bio-rad)). Swi4-3V5 or untagged Swi4, Swi6, and Mbp1 proteins were all transferred onto 0.45 µm nitrocellulose membranes using a PROTEAN Tetra tank (BioRAD) filled with 25 mM Tris, 192 mM glycine, and 7.5% methanol. All blots were incubated at room temperature with Odyssey Blocking Buffer (PBS; LI-COR Biosciences).

Immunoblotting for Cln1-3V5, Cln2-3V5, and GFP-Ime1 was performed as previously described in *Tresenrider et al., 2021*. Briefly, mouse α-V5 antibody (R960-25, Thermo Fisher) or mouse α-GFP antibody (632381, Takara) were diluted 1:2000 in Odyssey Blocking Buffer (PBS) (LI-COR Biosciences) with 0.01% Tween. Rabbit α-hexokinase (α-Hxk2) antibody (H2035, US Biological) was diluted to 1:20,000. Secondary antibodies used were α-mouse antibody conjugated to IRDye 800CW (926–32212, LI-COR Biosciences) and α-rabbit antibody conjugated to IRDye 680RD (926–68071, LI-COR Biosciences). Secondary antibodies were diluted to 1:20,000 in Odyssey Blocking Buffer (PBS) with 0.01% Tween.

For immunoblotting of Swi4, Swi6, and Mbp1. antibodies specific to each subunit were a generous gift from the Andrews and deBruin labs (*Andrews and Herskowitz, 1989*; *Harris et al., 2013*). α-Swi4, α-Swi6, α-Mbp1 antibodies were each diluted to 1:2000 in Odyssey Blocking Buffer (PBS) (LI-COR Biosciences) with 0.01% Tween. Secondary antibodies included a α-rabbit antibody conjugated to IRDye 800CW (926–32213, LI-COR Biosciences) and a α-rabbit antibody conjugated to IRDye 680RD (926–68071, LI-COR Biosciences).

Odyssey system (LI-COR Biosciences) was used to image the blots, and Image Studio Lite (LI-COR Biosciences) was used for image quantification.

## Northern (RNA) blotting

For each blot, 10 µg of total RNA was dried in a Savant Speed Vac (SPD111V). RNA was then resuspended and denatured in glyoxal/DMSO mix (1 M deionized glyoxal, 50% v/v DMSO, 10 mM sodium phosphate (NaPi) buffer [pH 6.8]) at 70 °C for 10 min. RNA sample was loaded into an agarose gel (1.1% [w/v] agarose in 0.01 M NaPi buffer) with loading dye (10% v/v glycerol, 2 mM NaPi buffer [pH 6.8],~0.25% w/v xylene cyanol, and orange G) and ran for 3 h at 100 V with a Variable Speed Pump (BioRad) to circulate buffer during the entire gel run.

RNA was transferred overnight to nylon membrane (Hybond-N+ [GE]) in SSC. Membrane was crosslinked using a Stratalinker UV Crosslinker (Stratagene). Ribosomal RNA (rRNA) bands were visualized with methylene blue staining and imaged on a Gel Doc XR +Molecular Imager with Image Lab software (BioRad).

Probe templates containing the T7 promoter were amplified using PCR. PCR product was concentrated with MinElute Spin Columns (Qiagen) and then used for in vitro transcription to generate a strand-specific RNA probe using a MaxiScript T7 Kit (Invitrogen) according to the manufacturer's instructions, except cold UTP was replaced with α-P32 labeled UTP (PerkinElmer). Excess nucleotides were removed with NucAway Spin Columns (Invitrogen). Blots were blocked in ULTRAhyb Ultrasensitive Hybridization Buffer (Invitrogen) for 1 hr and then incubated with the α-P32 labeled probe overnight at 68 °C. Blots were then washed twice with low stringency wash buffer (2 X SSC, 0.1% SDS) for 10 min and then washed twice with high stringency wash buffer (0.1 X SSC, 0.1% SDS) for 15 min. Blots were then exposed overnight on a storage phosphor screen (Molecular Dynamics) and then imaged on a Typhoon phosphor-imaging system.

## Fluorescence microscopy

For time course imaging of cells expressing *GFP-IME1*, *SWI4-mCherry*, or *WHI5-mCherry*, 500 µL of meiotic culture was fixed with a final concentration of 3.7% formaldehyde (v/v) at room temperature for 15 min. Cells were then washed in 1 ml of 100 mM potassium phosphate [pH 6.4] and stored at 4 °C in 20 µl of KPi Sorbitol solution overnight (100 mM potassium phosphate [pH 7.5], 1.2 M sorbitol). Cells were mounted on a slide and imaged using DeltaVision Elite wide-field fluorescence microscope (GE Healthcare) with a 60 x/1.516 oil immersion objective. Deconvolution of images was done with softWoRx imaging software (GE Life Sciences).

For live-cell imaging, cells at $OD_{600}$ of 1.85 in conditioned SPO (filter-sterilized SPO culture after ~5 hr in 30 °C) were sonicated and transferred to a concanavalin A (Sigma) treated 96-well clear, flat bottom plate (Corning). Four z positions (2 µm step size) were acquired per XY position. Acquisition was performed in a temperature-controlled chamber at 30 °C. Please refer to *Supplementary file 7* for acquisition settings.

## Image quantification

All image analysis was performed with FIJI (*Schindelin et al., 2012*). Maximum z-projection are shown in figures and were modified using linear brightness and contrast adjustments in FIJI.

To quantify localization of GFP-Ime1 and Rec8-GFP, z-slices containing the nucleus were selected using the Htb1-mCherry signal. Max projection was created from these slices and GFP-Ime1 was scored double-blinded as nuclear or not nuclear.

For mean nuclear intensity of GFP-Ime1 and Swi4-mCherry, an individual z-slice containing the nucleus was selected and nuclear mask was generated using the Htb1-mCherry signal. The nuclear mask was then used to quantify the mean nuclear intensity of GFP-Ime1 signal.

## Single molecule RNA FISH

smFISH was performed and quantified as previously described in *Chen et al., 2018*. All probes were ordered from (Biosearch Technologies). Unique region of *SWI4^LUTI^* was visualized by twenty-eight 20-mer oligonucleotide probes coupled to CAL fluor Red 590. Thirty-eight 20-mer probes coupled to Quasar 670 dye targeted to *SWI4^canon^*. Approximate;y 4 OD unit of cells were fixed in final 3% formaldehyde (v/v) and incubated at room temperature for 20 min. Fixed samples were moved to 4 °C to continue fixing overnight. Cells were washed three times in cold Buffer B (0.1 M potassium phosphate [pH 7.5], 1.2 M sorbitol) and resuspended in digestion buffer (Buffer B, 200 mM Vanadyl ribonucleoside complex [VRC from NEB], zymolyse [zymolase 100T, MP Biomedicals]). Cells were digested at

30 °C for 20 min and then gently washed with 1 mL of cold Buffer B and resuspended in 1 mL of 70% ethanol for 3.5–5 hr.

Cells were then incubated in 1 mL of 10% formamide wash buffer (10% formamide, 2 X SSC) at room temperature for 15 min. For hybridization, each probe set was added (final concentration of 500 nM) to 20 mM VRC and hybridization buffer (1% Dextran sulfate [EMD Millipore], 1 mg/mL *E. coli* tRNA [Sigma], 2 mM VRC, 0.2 mg/mL BSA, 1 X SSC, 10% formamide in nuclease-free water). Hybridization was done overnight at 30 °C. Samples were incubated in the dark for 30 min at 30 °C in 1 mL of 10% formamide wash buffer. Buffer was then removed, and cells were stained with DAPI and resuspended in glucose-oxygen-scavenging buffer or GLOX buffer (10 mM Tris [pH 8.0], 2 x SSC, 0.4% glucose) without enzymes. Before imaging, GLOX solution with enzyme (1% v/v catalase, 1% v/v glucose oxidase (Sigma), 2 mM Trolox (Sigma)) was added to sample.

Images were acquired with the DeltaVision microscope as described in previous section with filters: TRITC (EX542/27, EM597/45) for CAL Fluor Red 590 and CY5 (EX632/22, EM679/34) for Quasar 670. Series of z-stacks (15–25 slices) were acquired with a step size of 0.2 µm.

Matlab script (*Chen et al., 2017*; *Chen et al., 2018*) was run to quantify FISH spots with max intensity projection of z-stacks. The same 'signal' and 'SNR' thresholds were applied to all the images within a replicate.

## Acknowledgements

We thank Gloria Brar, Nick Ingolia, Rebecca Heald, Kathlyeen Ryan, Jingxun Chen, Anthony Harris, Andrea Higdon, Grant King, Jessica Leslie, Kate Morse, Emily Powers, Cyrus Ruediger, Tina Sing, and Ben Styler for suggestions and comments on this manuscript, David McSwiggen and Yuichiro Iwamoto for technical support, all members of the Ünal and Brar labs and Jeremy Thorner for valuable discussions. This work is supported by funds from the National Institutes of Health (R01 GM140005) and Astera Institute to EÜ and a National Institutes of Health Traineeship (T32 GM007232) to AJS.

## Additional information

### Competing interests

Elçin Ünal: Reviewing editor, *eLife*. The other authors declare that no competing interests exist.

### Funding

| Funder | Grant reference number | Author |
| --- | --- | --- |
| National Institutes of Health | R01 GM140005 | Elçin Ünal |
| Astera Institute | | Elçin Ünal |
| National Institutes of Health | T32 GM007232 | Amanda J Su |

The funders had no role in study design, data collection and interpretation, or the decision to submit the work for publication.

### Author contributions

Elçin Ünal, Conceptualization, Resources, Supervision, Funding acquisition, Investigation, Methodology, Writing – original draft, Project administration, Writing – review and editing; Amanda J Su, Conceptualization, Formal analysis, Supervision, Investigation, Methodology, Writing – original draft, Writing – review and editing; Siri C Yendluri, Formal analysis, Investigation, Methodology

### Author ORCIDs

Elçin Ünal ⓘ http://orcid.org/0000-0002-6768-609X
Amanda J Su ⓘ http://orcid.org/0009-0002-4203-1713

Joint Public Review: https://doi.org/10.7554/eLife.90425.3.sa1

Author Response https://doi.org/10.7554/eLife.90425.3.sa2

## Additional files

### Supplementary files

• Supplementary file 1. Differentially expressed genes (DESeq2) plotted in *Figure 2F* for *pATG8-SWI4* (UB22226) vs wild type (UB22199).

• Supplementary file 2. Sporulation efficiency of cells at 24 hr in SPO media for Rec8-GFP (UB32085), Rec8-GFP; *pATG8-SWI4* (UB32089). 200 cells counted per strain. Second table is sporulation efficiency of cells at 24 hr in SPO media for the following genotypes: wild type (UB22199), *pATG8-CLN1* (UB32820), *pATG8-CLN2* (UB25959), *pCUP-GFP-IME1* (UB34641), *pCUP1-GFP-IME1*; *pATG8-CLN2* (UB35057), *PUS1-αGFP* (UB35593), *PUS1-αGFP*; *pATG8-CLN2* (UB35982), *UME6-αGFP* (UB35300), and *UME6-αGFP*; *pATG8-CLN2* (UB35177).

• Supplementary file 3. Differentially expressed genes (DESeq2) plotted in *Figure 6A, C and D*. First tab is *WHI5-AA*; *ΔLUTI* vs. wild type. Second tab is *WHI5-AA* vs. wild type. Third tab is *ΔLUTI* versus wild type.

• Supplementary file 4. The genotypes of the strains used in this study.

• Supplementary file 5. Plasmids used in this study.

• Supplementary file 6. Primers used for qPCR and RNA blotting in this study.

• Supplementary file 7. Image acquisition information.

• MDAR checklist

### Data availability

All materials used in this study are available upon request from the corresponding author. Sequencing data generated in this study are available at NCBI GEO under the accession ID: GSE225963. The custom code used for the analysis is available in the following code repository: https://github.com/elifesciences-publications/Chen_Tresenrider_et_al_2017 (*McSwiggen, 2017*).

The following dataset was generated:

| Author(s) | Year | Dataset title | Dataset URL | Database and Identifier |
|---|---|---|---|---|
| Su AJ, Yendluri SC, Unal E | 2023 | LUTI-dependent Downregulation of a Cell Cycle Transcription Factor is Key for Timely Meiotic Entry | http://www.ncbi.nlm.nih.gov/geo/query/acc.cgi?acc=GSE225963 | NCBI Gene Expression Omnibus, GSE225963 |

The following previously published dataset was used:

| Author(s) | Year | Dataset title | Dataset URL | Database and Identifier |
|---|---|---|---|---|
| Brar GA, Yassour M, Friedman N, Regev A, Ingolia NT, Weissman JS | 2012 | High-resolution view of the yeast meiotic program revealed by ribosome profiling | https://www.ncbi.nlm.nih.gov/geo/query/acc.cgi?acc=GSE34082 | NCBI Gene Expression Omnibus, GSE34082 |

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
