## [Editor Report · eLife assessment]

This study highlights several **important** regulatory pathways that contribute to the control of entry into meiosis by turning down mitotic functions. Central to this regulation is the control of Swi4 level and activity, and **convincing** overexpression experiments identify downstream effectors of Swi4.

---

## [Referee Report · Joint Public Review]

The manuscript highlights a mechanistic insight into meiotic initiation in budding yeast. In this study, the authors analyzed the genetic link between the mitotic cell cycle regulator SBF (the Swi4-Swi6 complex) and a meiosis inducing regulator Ime1 in the context of meiotic initiation. The authors' comprehensive analyses with cytology, imaging, RNA-seq using mutant strains lead to the conclusion that Swi4 levels regulates Ime1-Ume6 interaction to activate expression of early meiosis genes for meiotic initiation.

The authors first show a down regulation of Swi4 at the protein level upon meiosis entry and then investigate downstream consequences. This study reveals several regulations: (1) Mutations in CLN1 and 2, which are targets of Swi4, allow rescuing the delay in meiotic entry observed when Swi4 is overexpressed; (2) Ime1 activity is antigonized by Swi4, and more specifically its interaction with Ume6. (3) Expression of SWI4 is regulated by LUTI-based transcription at the SWI4 locus that impedes expression of canonical SWI4 transcripts (4) The expression of SWI4 LUTI is likely negatively regulated by the Ime1-Ume6 complex (5) Whi5 restrict SBF activity during meiotic entry, thereby ensuring Cyclin repression.

The important implication in this paper is that meiotic initiation is regulated by the balance of mitotic cell cycle regulator and meiosis-specific transcription factor.

---

## [Author Response]

The following is the authors’ response to the original reviews.

**Reviewer #1 (Public Review):**
Su et al propose the existence of two mechanisms repressing SBF activity during entry into meiosis in budding yeast. First, a decrease in Swi4 protein levels by a LUTI-dependent mechanism where Ime1 would act closing a negative feedback loop. Second, the sustained presence of Whi5 would contribute to maintaining SBF inhibited under sporulation conditions. The article is clearly written and the experimental approaches used are adequate to the aims of this work. The results obtained are in line with the conclusions reached by the authors but, in my view, they could also be explained by the existing literature and, hence, would not represent a major advance in the field of meiosis regulation.

We respectfully disagree with the reviewer about their comment that this work can be explained by the existing literature. First, while SWI4LUTI has been previously identified in meiotic cells along with ~ 380 LUTIs, the biological purpose of these alternative mRNA isoforms and their effect on cellular physiology still remain largely unknown. Our manuscript clarifies this gap in understanding for SWI4LUTI. Loss of SWI4LUTI contributes to dysregulation of meiotic entry and does so by failing to properly repress the known inhibitors of meiotic entry, the CLNs. Furthermore, even though Cln1 and Cln2 have been previously shown to antagonize meiosis, the mechanisms that restrict their activity was unclear prior to our study.

We recognize work done by others demonstrating Whi5-dependent repression of SBF during mitotic G1/S transition (De Bruin et al., 2004; Costanzo et al., 2004). We further examined Whi5’s involvement during meiotic entry and found that it acts in conjunction with the LUTI-based mechanism to restrict SBF activity. Combined loss of both mechanisms results in the increased expression of G1 cyclins, decreased expression of early meiotic genes, and a delay in meiotic entry (Figure 6). Neither mechanism was previously known to regulate meiotic entry. Our study not only adds to our broader understanding of gene regulation during meiosis but also raises additional questions regarding how LUTIs regulate gene expression and function.

Regarding the first mechanism, Fig 1 shows that Swi4 decreases very little after 1-2h in sporulation medium, whereas G1-cyclin expression is strongly repressed very rapidly under these conditions (panel D and work by others). This fact dampens the functional relevance of Swi4 downregulation as a causal agent of G1 cyclin repression.

Reviewer 1 expresses concern for the observation that by 2 h in sporulation media there is a 32% decrease in Swi4-3V5 protein abundance compared to 0 h in SPO. This is consistent with the range of protein level decrease typically accomplished by LUTI-based gene regulation (Chen et al., 2017; Chia et al., 2017; Tresenrider et al., 2021), and while it is a modest reduction, it is consistent across replicates. Furthermore, we don’t make the argument that reduction in Swi4 levels alone is the sole regulator of G1 cyclin levels. In fact, we report that in addition to Swi4 downregulation, Whi5 also functions to restrict SBF activity during meiotic entry, thereby ensuring G1 cyclin repression.

In addition, the LUTI-deficient SWI4 mutant does not cause any noticeable relief in CLN2 repression, arguing against the relevance of this mechanism in the repression of G1-cyclin transcription during entry into meiosis. The authors propose a second mechanism where Whi5 would maintain SBF inactive under sporulation conditions. The role of Whi5 as a negative regulator of the SBF regulon is well known. On the other hand, the double WHI5-AA SWI4-dLUTI mutant does not upregulate CLN2, the G1 cyclin with the strongest negative effect on sporulation, raising serious doubts on the functional relevance of this backup mechanism during entry into meiosis.

Due to replicate variance, CLN2 did not make the cut by our mRNA-seq data analysis as a significant hit. To address reviewer 1’s final point we opted for the “gold standard” of reverse transcription coupled with qPCR to measure CLN2 transcript levels in the double mutant ∆LUTI; WHI5-AA and the wild-type control. This revealed that CLN2 levels were significantly increased in the double mutant compared to wild type at 2 h in SPO (Author Response Image 1, *, p = 0.0288, two-tailed t-test).

**Author response image 1. sa2fig1:** Wild type (UB22199) and ∆LUTI;WHI5-AA (UB25428) cells were collected to perform RT-qPCR for CLN2 transcript abundance. Transcript abundance was quantified using primer sets specific for each respective gene from three technical replicates for each biological replicate. Quantification was performed in reference to PFY1 and then normalized to wild-type control. FC=fold change. Experiments were performed twice using biological replicates, mean value plotted with range. Differences in wild type versus ∆LUTI; WHI5-AA transcript levels compared with a two-tailed t-test (*, p = 0.0288)

**Reviewer #2 (Public Review):**
Summary:The manuscript highlights a mechanistic insight into meiotic initiation in budding yeast. In this study, the authors addressed a genetic link between mitotic cell cycle regulator SBF (the Swi4-Swi6 complex) and a meiosis inducing regulator Ime1 in the context of meiotic initiation. The authors' comprehensive analyses with cytology, imaging, RNA-seq using mutant strains lead the authors to conclude that Swi4 levels regulates Ime1-Ume6 interaction to activate expression of early meiosis genes for meiotic initiation. The major findings in this paper are that (1) the higher level of Swi4, a subunit of SBF transcription factor for mitotic cell cycle regulation, is the limiting factor for mitosis-to-meiosis transition; (2) G1 cyclins (Cln1, Cln2), that are expressed under SBF, inhibit Ime1-Ume6 interaction under overexpression of SWI4, which consequently leads to downregulation of early meiosis genes; (3) expression of SWI4 is regulated by LUTI-based transcription in the SWI4 locus that impedes expression of canonical SWI4 transcripts; (4) expression of SWI4 LUTI is likely negatively regulated by Ime1; (5) Action of Swi4 is negatively regulated by Whi5 (homologous to Rb)-mediated inhibition of SBF, which is required for meiotic initiation. Thus, the authors proposed that meiotic initiation is regulated under the balance of mitotic cell cycle regulator SBF and meiosis-specific transcription factor Ime1.Strengths:The most significant implication in their paper is that meiotic initiation is regulated under the balance of mitotic cell cycle regulator and meiosis-specific transcription factor. This finding will provide a mechanistic insight in initiation of meiosis not only into the budding yeast also into mammals. The manuscript is overall well written, logically presented and raises several insights into meiotic initiation in budding yeast. Therefore, the manuscript should be open for the field. I would like to raise the following concerns, though they are not mandatory to address. However, it would strengthen their claims if the authors could technically address and revise the manuscript by putting more comprehensive discussion.Weaknesses:The authors showed that increased expression of the SBF targets, and reciprocal decrease in expression of meiotic genes upon SWI4 overexpression at 2 h in SPO (Figure 2F). However, IME1 was not found as a DEG in Supplemental Table 1. Meanwhile, IME1 transcript level was decreased at 2 h SPO condition in pATG8-CLN2 cells in Fig S4C.Now this reviewer still wonders with confusion whether expression of IME1 transcripts per se is directly or in directly suppressed under SBF-activated gene expression program at 2 h SPO in pATG8-SWI4 and pATG8-CLN2 cells. This reviewer wonders how Fig S4C data reconciles with the model summarized in Fig 6F.One interpretation could be that persistent overexpression of G1 cyclin caused active mitotic cell cycle, and consequently delayed exit from mitotic cell cycle, which may have given rise to an apparent reduction of cell population that was expressing IME1. For readers to better understand, it would be better to explain comprehensively this issue in the main text.

We believe there was an oversight here. In supplemental table 1, IME1 expression is reported as significantly decreased. The volcano plot shown below also highlights this change (Author response image 2).

**Author response image 2. sa2fig2:** Volcano plot of DE-Seq2 analysis for ∆LUTI;WHI5-AA versus wild type. Dashed line indicates padj (p value) = 0.05. Analysis was performed using mRNA-seq from two biological replicates. Wild type (UB22199) and ∆LUTI;WHI5-AA (UB25428) cells were collected at 2 h in SPO. SBF targets (pink) (Iyer et al., 2001) and early meiotic genes (blue) defined by (Brar et al., 2012). Darker pink or darker blue, labeled dots are well studied targets in either gene set list.

The % of cells with nuclear Ime1 was much reduced in pATG8-CLN2 cells (Fig 2B) than in pATG8-SWI4 cells (Fig 4C). Is the Ime1 protein level comparable or different between pATG8-CLN2 strain and pATG8-SWI4 strain? Since it is difficult to compare the quantifications of Ime1 levels in Fig S1D and Fig S4B, it would be better to comparably show the Ime1 protein levels in pATG8-CLN2 and pATG8-SWI4 strains.Further, it is uncertain how pATG8-CLN2 cells mimics the phenotype of pATG8-SWI4 cells in terms of meiotic entry. It would be nice if the authors could show RNA-seq of pATG8-CLN2/WT and/or quantification of the % of cells that enter meiosis in pATG8-CLN2.

Analyzing bulk Ime1 protein levels across a population of cells (Author response image 3) reveals that overexpression of CLN2 causes a more severe decrease in Ime1 levels than overexpression of SWI4. This is consistent with our observation that pATG8-CLN2 has a more severe impact on meiotic entry than pATG8-SWI4. The higher CLN2 levels (Author response image 4) likely accounts for the observed difference in severity of phenotype between the two mutants.

**Author response image 3. sa2fig3:** Samples from strain wild type (UB22199), pATG8-SWI4 (UB2226), pATG8-CLN2 (UB25959) and were collected between 0-4 hours (h) in sporulation medium (SPO) and immunoblots were performed using α-GFP. Hxk2 was used a loading control.

**Author response image 4. sa2fig4:** Wild type (UB22199), pATG8-SWI4 (UB2226), pATG8-CLN2 (UB25959) cells were collected to perform RT-qPCR for CLN2 transcript abundance. Quantification was performed in reference to PFY1 and then normalized to wild-type control. FC=fold change.

The authors stated that reduced Ime1-Ume6 interaction is a primary cause of meiotic entry defect by CLN2 overexpression (Line 320-322, Fig 4J-L). This data is convincing. However, the authors also showed that GFP-Ime1 protein level was decreased compared to WT in pATG8-CLN2 cells by WB (Fig S4A).

Compared to wild type, pATG8-CLN2 cells have lower levels of Ime1. Consequently, reviewer 2 suggests that this reduction may be responsible for the observed meiotic defect. However, we tested this possibility and found it not to be the primary cause of the meiotic defect in pATG8-CLN2 cells. As shown in Figure S4A, when IME1 was overexpressed from the pCUP1 promoter, Ime1 protein levels were similar between wild-type and pATG8-CLN2 cells. Despite this similarity, we still observed a decrease in nuclear Ime1 (Figure 4F) and no rescue in sporulation (Figure 4A). Therefore, the reduction in Ime1 protein levels alone cannot explain the meiotic defect caused by CLN2 overexpression.

Further, GFP-Ime1 signals were overall undetectable through nuclei and cytosol in pATG8-CLN2 cells (Fig 4B), and accordingly cells with nuclear Ime1 were reduced (Fig 4C). Although the authors raised a possibility that the meiotic entry defect in the pATG8-CLN2 mutant arises from downregulation of IME1 expression (Line 282-283), causal relationship between meiotic entry defect and CLN2 overexpression is still not clear.

As reviewer 2 comments, we initially considered the possibility that meiotic entry defect induced by CLN2 overexpression could be attributed to decreased IME1 expression. However, in the following paragraph in the manuscript, we demonstrate equalizing IME1 transcript levels using the pCUP1-IME1 allele does not rescue the meiotic defect caused by CLN2 overexpression. Consequently, we conclude that the decrease in IME1 transcript levels alone cannot explain the meiotic defect caused by increased CLN2 levels.

Is the Ime1 protein level reduced in the pATG8-CLN2;UME6-⍺GFP strain compared to WT? It would be better to comparably show the Ime1 protein levels in the pATG8-CLN2 strain and the pATG8-CLN2;UME6-⍺GFP strain by WB. Also, it would be nice if the authors could show quantification of the % of cells that enter meiosis in the pATG8-CLN2;UME6-⍺GFP strain to see how and whether artificial tethering of Ime1 to Ume6 rescued normal meiosis program rather than simply showing % sporulation in Fig4A.

We do not agree with the suggestion to compare the pATG8-CLN2;UME6-⍺GFP with wild type as the kinetics of meiosis is rather different. The more appropriate comparison is UME6-⍺GFP and pATG8-CLN2;UME6-⍺GFP which shows GFP-Ime1 bulk protein levels are slightly lower (Author response image 5). However, when we use a more sensitive measurement of meiotic entry through the nuclear accumulation of Ime1 in single cells, as illustrated in Figure 4L, it becomes evident that the Ume6-Ime1 tether is capable of restoring nuclear Ime1 levels, even in the presence of CLN2 overexpression. Given that these cells exhibited wild type levels of nuclear Ime1 and underwent sporulation after 24 hours, we make the fair assumption that they have successfully initiated the meiotic program.

**Author response image 5. sa2fig5:** Wild type (UB22199), pATG8-SWI4 (UB35106), UME6-⍺GFP (UB35300), and UME6-⍺GFP; pATG8-CLN2 (UB35177) cells collected between 0-3 hours (h) in sporulation medium (SPO) and immunoblots were performed using α-GFP. Hxk2 was used a loading control

The authors showed Ume6 binding at the SWI4LUTI promoter (Figure 5K). However, since Ume6 forms a repressive form with Rpd3 and Sin3a and binds to target genes independently of Ime1, Ume6 binding at the SWI4LUTI promoter bind does not necessarily represent Ime1-Ume6 binding there. Instead, it would be better to show Ime1 ChIP-seq at the SWI4LUTI promoter.

We agree with reviewer 2 that Ime1 ChIP would be the ideal measurement. Unfortunately, this has proved to be technically challenging. To address this limitation, we utilized a published Ume6 ChIP-seq dataset along with a published UME6-T99N RNA-seq dataset. Cells carrying the UME6-T99N allele are unable to induce the expression of early meiotic transcripts due to lack of Ime1 binding to Ume6 (Bowdish et al., 1995). Accordingly, RNA-seq analysis should reveal whether or not the LUTIs identified by Ume6 ChIP are indeed regulated by Ime1-Ume6 during meiosis. For SWI4LUTI, this is exactly what we observe. Not only is there Ume6 binding at the SWI4LUTI promoter (Figure 5K), but there is also a significant decrease in SWI4LUTI expression in UME6-T99N cells under meiotic conditions (Figure S5). Based on these data, we conclude that the Ime1-Ume6 complex is responsible for regulating SWI4LUTI expression during meiosis.

The authors showed ∆LUTI mutant and WHI5-AA mutant did not significantly change the expression of SBF targets nor early meiotic genes relative to wildtype (Figure 6A, C). Accordingly, they concluded that LUTI- or Whi5-based repression of SBF alone was not sufficient to cause a delay in meiotic entry (Line451-452), and perturbation of both pathways led to a significant delay in meiotic entry (Figure 6E). This reviewer wonders whether Ime1 expression level and nuclear localization of Ime1 was normal in ∆LUTI mutant and WHI5-AA mutant.

Based on our observations in Figure 4, Ime1 protein and expression levels were not reliable indicators of meiotic entry. Consequently, we opted for a more downstream and functionally relevant measure of meiotic entry, which involved time-lapse fluorescence imaging of Rec8, an Ime1 target.

**Reviewer #1 (Recommendations For The Authors):**
The authors would like to mention previous work showing that G1-cyclin overexpression decreases the expression and nuclear accumulation of Ime1 (Colomina et al 1999 EMBO J 18:320). In this work, the interaction between Ime1 and Ume6 had been found to be resistant to G1-cyclin expression, arguing against a direct effect on the recruitment of Ime1 at meiotic promoters. Alternatively, differences in the experimental approaches used could be discussed to explain this apparent discrepancy.

To clarify, in the paper that reviewer 1 is referring to (Colomina et al., 1999), the authors determine that the interaction between Ime1 and Ume6 is regulated by the presence of a non-fermentable carbon source. Additional work by others reveals that Ime1 undergoes phosphorylation by the protein kinases Rim11 and Rim15, promoting its nuclear localization and enabling interaction with Ume6 (Vidan and Mitchell, 1997; Pnueli et al., 2004; Malathi et al., 1999, 1997). Furthermore, both Rim11 and Rim15 kinase activities are inhibited by the presence of glucose via the PKA pathway (Pedruzzi et al., 2003; Rubin-Bejerano et al., 2004; Vidan and Mitchell, 1997). Accordingly, the elimination of cyclins in the presence of a non-fermentable carbon source (glucose) in (Colomina et al., 1999) is unlikely to result in an interaction between Ime1 and Ume6, as Rim11 and Rim15 remain repressed. Removal of cyclins in acetate does not further increase Ime1-Ume6 interaction leading the authors to conclude that G1 cyclins do not block Ime1 function through its interaction with Ume6. This work however uses loss of function (removal of G1 cyclins) to study the G1 cyclins’ effect on Ime1-Ume6 interaction while using timepoints that are well beyond meiotic entry. Additionally, Ime1-Ume6 interaction is being tested using yeast-two hybrid analysis with just the proposed interaction domain of Ime1 (amino acids 270-360). Therefore, the interpretation that G1 cyclins are dispensable for regulating the interaction between Ime1 and Ume6 is unclear from this work alone.

There are many differences that can explain the discrepancy between our work and (Colomina et al., 1999). Our work uses increased expression of cyclins during meiotic entry. Additionally, in our study, we collected timepoints to measure meiotic entry (2 h in SPO) and sporulation (gamete formation) efficiency (24 h in SPO). Finally, we are using the endogenous, full length Ime1. These differences could very well explain the discrepancy with previous work. Lastly, in our discussion we acknowledge the lack of CDK consensus phosphorylation sites on Ime1. Therefore, it is most likely that G1 cyclins are not directly phosphorylating Ime1 and that other factors like Rim11 and Rim15 could be direct targets of the G1 cyclins, considering their involvement in the phosphorylation of Ime1-Ume6, as well as their role in regulating Ime1 localization and its interaction with Ume6. We have included these points in the revised manuscript (lines 547-551).

**Reviewer #2 (Recommendations For The Authors):**
This reviewer thinks that the findings in this paper are of general interest to meiosis field and help understanding the mechanism of meiotic initiation in mammals. The way of the current manuscript seems to be written for limited budding yeast scientists, and should not limited to the interest by the budding yeast scientists. Thus, it would be better to discuss more about what is known about the mechanism of initiation of meiosis not only in budding yeast but also in other species to share their finding to more broad scientists using other organisms.

We appreciate reviewer 2’s comment and have added more discussion about the parallels between yeast and mammalian systems in meiotic initiation (lines 613-624).

**Reviewer #3 (Recommendations For The Authors):**
The effect of overexpression of Swi4 is tested for MI and MII (Fig1F): this is a very indirect readout of meiotic entry. The authors could present Rec8 localization (Fig2I) at this stage. However, this is still a superficial description of the meiotic phenotype: is the phenotype only a delay or is the meiotic prophase altered. It is specifically important to analyse this in more detail to answer whether the overexpression of Swi4 leads to an identical phenotype to the one of CLN2. Also the comparison between overexpression of Swi4 and Cln2 is difficult to evaluate: what is the level of CLN2 when SwI4 is overexpressed compared to CLN2 overexpression. The percentage of nuclear Ime1 is 50% vs 5% when Swi4 or Cln2 are overexpressed. What is the interpretation? What are the levels of Ime1? (Y axis of quantifications not comparable, see also comment for Fig5F,H)

CLN2 is expressed at a much higher level in pATG8-CLN2 cells relative to pATG8-SWI4 (Author Response Image 4). Therefore, we don’t expect identical phenotypes, but rather a more severe deficiency in meiotic entry upon CLN2 overexpression. The key experiment that establishes causality between SWI4 and CLNs is reported in Figure 3, where deletion of either CLN1 or CLN2 rescues the meiotic entry delay exerted by SWI4 overexpression.

Fig3EF: What is the phenotype of Cln1 and Cln2 without overexpression of Swi4?

Meiotic entry is not faster in cln1∆ or cln2∆ cells compared to wild-type. We included these data in Supplemental Figure 3 and made the relevant changes in the manuscript (lines 257-261).

Fig4F: Need a control with CLN2 overexpression only.

A control with only CLN2 overexpression (pATG8-CLN2) is not appropriate since these meiotic time course experiments are synchronized using the pCUP1-IME1 allele. It would be a misleading comparison since the two meiosis would have different kinetics. Figure 4F reports that despite similar IME1 transcript levels and Ime1 protein levels, CLN2 overexpressing cells still have reduced nuclear Ime1. Since side-by-side comparison of pATG8-CLN2 and pCUP1-IME1 is not possible, we chose to measure sporulation efficiency at 24 h in Figure 4A. These data together suggest that elevated IME1 transcript and protein levels cannot rescue the defects associated with increased CLN2 expression.

Fig5E: in wild type, by Northern blot, Swi4canon level is increasing during meiosis, not decreasing?, whereas protein level is decreasing, what is the interpretation?

Northern data is less quantitative than smFISH, which show that SWI4canon transcript levels are significantly lower in meiosis compared to vegetative cells (Figure 5D). We also note that the Northern blot data were acquired from unsynchronized meiotic cells and could have additional limitations based on the population-based nature of the assay. Finally, additional analysis of a transcript leader sequencing (TL-seq) dataset from synchronized cells (Tresenrider et al., 2021) further confirms the decrease in SWI4canon transcript levels upon meiotic entry. (Author response image 6).

**Author response image 6. sa2fig6:** TL-seq data from Tresenrider et al. 2021 visualized on IGV at the SWI4 locus. Two timepoints are plotted including premeiotic before IME1 induction (pink) and meiotic prophase or after IME1 induction (blue).

Fig5F, H. This quantification needs duplicates for validation.

Replicates are submitted for every blot in this paper to eLIFE.It can be found in the shared Dropbox folder to the editors (named Raw-blots-for-eLIFE).

Fig5F, H. Why are the wild type values so different?

The immunoblotting done between Figure 5F and Figure 5H are on separate blots and therefore should not be compared. Additionally, these values are not absolute measurements of wild type values of Swi4-3V5 and therefore we should not expect them to be the same. Any comparisons done of relative amounts of Swi4-3V5 are always done on the same blot and normalized to a loading control, hexokinase.

FigS5: What is the effect of the Ume6-T99N on Swi4 protein level and on meiotic entry?Is the backup mechanism proposed active?

We haven’t measured Swi4 protein levels in the UME6-T99N background but given that this mutation is known to disrupt the interaction between Ime1 and Ume6, we expect a similar trend to that reported in Figure 5I (pCUP1-IME1 uninduced).

What is the evidence that Swi4/6 is a E2F homolog? What is the homology at the protein level?

While there is no sequence homology between SBF and E2F there is remarkable similarity between metazoans and yeast in terms of the regulation of the G1/S transition (reviewed in Bertoli et al., 2013). E2F and SBF are both repressed before the G1/S transition by the inhibitors Rb and Whi5, respectfully (Costanzo et al., 2004; De Bruin et al., 2004; Hasan et al., 2014). During G1/S transition, a cyclin dependent kinase phosphorylates and inactivates these inhibitors. We have carefully edited our language in the manuscript to “functional homology” instead of just “homology”.

FigS3 is missing

Each supplemental figure was matched to its corresponding main figure. In the original submission, we didn’t have Figure S3. However, the revised manuscript now contains FigS3.

Bertoli, C., J.M. Skotheim, and R.A.M. De Bruin. 2013. Control of cell cycle transcription during G1 and S phases. Nat. Rev. Mol. Cell Biol. 14:518–528. doi:10.1038/nrm3629.

Bowdish, K.S., H.E. Yuan, and A.P. Mitchell. 1995. Positive control of yeast meiotic genes by the negative regulator UME6. Mol. Cell. Biol. 15:2955–2961. doi:10.1128/mcb.15.6.2955.

Brar, G.A., M. Yassour, N. Friedman, A. Regev, N.T. Ingolia, and J.S. Weissman. 2012. High-Resolution View of the Yeast Meiotic Program Revealed by Ribosome Profiling. Science (80-. ). 335:552–558. doi:10.1126/science.1215110.

De Bruin, R.A.M., W.H. McDonald, T.I. Kalashnikova, J. Yates, and C. Wittenberg. 2004. Cln3 activates G1-specific transcription via phosphorylation of the SBF bound repressor Whi5. Cell. 117:887–898. doi:10.1016/j.cell.2004.05.025.

Chen, J., A. Tresenrider, M. Chia, D.T. McSwiggen, G. Spedale, V. Jorgensen, H. Liao, F.J. Van Werven, and E. Ünal. 2017. Kinetochore inactivation by expression of a repressive mRNA. Elife. 6:1–31. doi:10.7554/eLife.27417.

Chia, M., A. Tresenrider, J. Chen, G. Spedale, V. Jorgensen, E. Ünal, and F.J. van Werven. 2017. Transcription of a 5’ extended mRNA isoform directs dynamic chromatin changes and interference of a downstream promoter. Elife. 6:1–23. doi:10.7554/eLife.27420.

Colomina, N., E. Garí, C. Gallego, E. Herrero, and M. Aldea. 1999. G1cyclins block the Ime1 pathway to make mitosis and meiosis incompatible in budding yeast. EMBO J. 18:320–329. doi:10.1093/emboj/18.2.320.

Costanzo, M., J.L. Nishikawa, X. Tang, J.S. Millman, O. Schub, K. Breitkreuz, D. Dewar, I. Rupes, B. Andrews, and M. Tyers. 2004. CDK activity antagonizes Whi5, an inhibitor of G1/S transcription in yeast. Cell. 117:899–913. doi:10.1016/j.cell.2004.05.024.

Hasan, M., S. Brocca, E. Sacco, M. Spinelli, P. Elena, L. Matteo, A. Lilia, and M. Vanoni. 2014. A comparative study of Whi5 and retinoblastoma proteins : from sequence and structure analysis to intracellular networks. 4:1–24. doi:10.3389/fphys.2013.00315.

Iyer, V.R., C.E. Horak, P.O. Brown, D. Botstein, V.R. Iyer, M. Snyder, and C.S. Scafe. 2001. Genomic binding sites of the yeast cell-cycle transcription factors SBF and MBF. Nature. 409:533–538. doi:10.1038/35054095.

Malathi, K., Y. Xiao, and A.P. Mitchell. 1997. Interaction of yeast repressor-activator protein Ume6p with glycogen synthase kinase 3 homolog Rim11p. Mol. Cell. Biol. 17:7230–7236. doi:10.1128/mcb.17.12.7230.

Malathi, K., Y. Xiao, and A.P. Mitchell. 1999. Catalytic roles of yeast GSK3β/shaggy homolog Rim11p in meiotic activation. Genetics. 153:1145–1152. doi:10.1093/genetics/153.3.1145.

Pedruzzi, I., F. Dubouloz, E. Cameroni, V. Wanke, J. Roosen, J. Winderickx, and C. De Virgilio. 2003. TOR and PKA Signaling Pathways Converge on the Protein Kinase Rim15 to Control Entry into G0. Mol. Cell. 12:1607–1613. doi:10.1016/S1097-2765(03)00485-4.

Pnueli, L., I. Edry, M. Cohen, and Y. Kassir. 2004. Glucose and Nitrogen Regulate the Switch from Histone Deacetylation to Acetylation for Expression of Early Meiosis-Specific Genes in Budding Yeast. Mol. Cell. Biol. 24:5197–5208. doi:10.1128/mcb.24.12.5197-5208.2004.

Rubin-Bejerano, I., S. Sagee, O. Friedman, L. Pnueli, and Y. Kassir. 2004. The In Vivo Activity of Ime1, the Key Transcriptional Activator of Meiosis-Specific Genes in *Saccharomyces cerevisiae*, Is Inhibited by the Cyclic AMP/Protein Kinase A Signal Pathway through the Glycogen Synthase Kinase 3- Homolog Rim11. Mol. Cell. Biol. 24:6967–6979. doi:10.1128/mcb.24.16.6967-6979.2004.

Tresenrider, A., K. Morse, V. Jorgensen, M. Chia, H. Liao, F.J. van Werven, and E. Ünal. 2021. Integrated genomic analysis reveals key features of long undecoded transcript isoform-based gene repression. Mol. Cell. 81:2231-2245.e11. doi:10.1016/j.molcel.2021.03.013.

Vidan, S., and A.P. Mitchell. 1997. Stimulation of yeast meiotic gene expression by the glucose-repressible protein kinase Rim15p. Mol. Cell. Biol. 17:2688–2697. doi:10.1128/mcb.17.5.2688.